# Dataset of Oil Slicks, Look-Alikes and Remarkable SAR Signatures Obtained from Sentinel-1 Data in the Eastern Mediterranean Sea

Yi-Jie Yang[1,2], Suman Singha[3,1], Ron Goldman[4], and Florian Schütte[5]

[1]Team SAR Oceanography, Remote Sensing Technology Institute, German Aerospace Center (DLR), Bremen, Germany
[2]Faculty of Mathematics and Natural Sciences, Kiel University, Kiel, Germany
[3]National Centre for Climate Research, Danish Meteorological Institute (DMI), Copenhagen, Denmark
[4]Israel Marine Data Center, Israel Oceanographic and Limnological Research (IOLR), Haifa, Israel
[5]GEOMAR Helmholtz Centre for Ocean Research Kiel, Kiel, Germany

**Correspondence:** Yi-Jie Yang (yi-jie.yang@dlr.de; yi-jie.yang@mailbox.org)

**Abstract.** Publicly available datasets for oil spill detection are scarce, making it difficult to compare the performance of different detection algorithms. To address this, this paper introduces a comprehensive labeled dataset of oil slicks, look-alikes, and other remarkable oceanic phenomena, derived from Sentinel-1 Synthetic Aperture Radar (SAR) products in the Eastern Mediterranean Sea in 2019. The dataset contains 3225 oil objects across 1365 image patches, along with an additional 2290 image patches featuring look-alikes or other phenomena. Data are available at https://doi.pangaea.de/10.1594/PANGAEA.980773 (Yang and Singha, 2025).

This dataset enables researchers to evaluate their oil spill detection models and compare performance with other studies. To facilitate this, the performance of an oil spill detector from a previous study on the dataset is provided as a baseline. In addition, to help the researchers better understand what phenomena their object detector might be confusing with oil slicks, the image patches without oil objects were sorted into several subgroups. On the other hand, for researchers looking to apply object detection models to oil slick detection but lacking a starting dataset, this dataset can serve as a valuable training resource. Beyond dataset presentation, this paper also explains the formation of different oceanic phenomena and their SAR signatures, supported by examples and supplementary materials. These insights help researchers from various backgrounds, such as remote sensing, oceanography, and machine learning, better understand the sources of SAR signatures.

## 1 Introduction

Spaceborne Synthetic Aperture Radar (SAR) has been widely applied to marine oil pollution detection. While airborne systems play an important role in emergency response due to their flexibility in time of deployment and choice of sensors, with the advantage of wide coverage and the ability to observe at night and through clouds, spaceborne SAR can be used to monitor oil spills on a regular basis and provide early warning (Brown and Fingas, 2005; Brekke and Solberg, 2005). Oil slicks typically appear as dark formations in SAR imagery, but other phenomena can also manifest dark formations that look similar and are difficult to distinguish. Distinguishing oil slicks from these *look-alikes* has long been a challenge.

With the increasing computational capacity and the growing number of accessible SAR scenes since the launch of Sentinel-1 in 2014, many recent studies have used deep learning-based methods to detect oil spills and distinguish them from look-alikes. However, due to the lack of publicly available oil spill datasets, most studies had to collect their own oil spill dataset for model training. Some studies collected images of major accidents, such as the DeepWater Horizon incident, the Hebei Spirit oil tanker collision, and accidents reported in the local news, and cropped the acquired SAR images into multiple image patches (Chen and Wang, 2022; Hasimoto-Beltran et al., 2023; Mahmoud et al., 2023). These studies employed a limited number of SAR scenes, with less than 36 images, which means that the training dataset may not reflect the differences in radar backscatter between different acquisitions well. Other studies relied on either manual inspection by the authors (Topouzelis and Psyllos, 2012; Amri et al., 2022; Chen et al., 2023) or local collaboration with other institutions or services (Konik and Bradtke, 2016; Cantorna et al., 2019; Zeng and Wang, 2020). These studies focused on the development of new oil spill detection models, but the use of different datasets made it difficult to compare the performance of different studies.

Recognizing the challenges posed by the absence of a comprehensive dataset, Krestenitis et al. (2019) published an oil spill detection dataset. It contains approximately 1000 Sentinel-1 SAR images with their corresponding ground truth masks, indicating five classes: oil spill, look-alikes, land, ship, and sea areas. The oil spills were reported by the European Maritime Safety Agency (EMSA) through the CleanSeaNet service. However, it is not openly available, the proposal is required to obtain the dataset. On the other hand, a recent work has contributed its training dataset to Zenodo, an open repository for datasets (Trujillo-Acatitla et al., 2024). The dataset includes oil spills reported by the National Oceanic and Atmospheric Administration (NOAA) and EMSA CleanSeaNet. There are 2850 image patches, half with oil spills inside, half without oil spills but with background or look-alikes inside. These two datasets offer pixel-wise classification of the images. However, in low-wind conditions or in areas where there are frequent look-alikes, there may be large areas of dark formations. For this reason, previous studies have proposed a two-step approach with coarse detection of oil slicks over a large area and refinement of the results to a pixel-wise level (Nieto-Hidalgo et al., 2018; Yang et al., 2024). This approach can theoretically increase efficiency, as the final detection is only performed if there are detections in the first step, and may therefore be helpful for a near real-time (NRT) monitoring system. Therefore, the dataset presented in this paper provides an object-based annotation of the oil spills, which aims to support research that follows the idea of this approach.

There are many different sources of look-alikes, such as low-wind areas, internal waves, upwelling, and biogenic films, that modify the ocean surface and introduce signatures in SAR imagery. It is important to understand which types of look-alikes the model cannot distinguish well, but this information is not available in previous datasets. Therefore, the published dataset includes image patches of oil slicks and other prominent marine signatures, stored in *oil set* and *no-oil set*, respectively. For the image patches in *no-oil set*, the K-Means unsupervised clustering algorithm was used to further sort the patches into several subgroups to help users better understand the performance of their algorithms on different sources of look-alikes. Note that K-Means categorizes each image as a whole under a specific cluster. In other words, unlike oil slicks in the *oil set*, which were labeled as objects, no annotation is attached to image patches in the *no-oil set*.

In addition, previous datasets used the reports from the existing service, but in many areas such services are not available. There is a lack of clear guidance on how to distinguish between different phenomena and what data can be used to support

them. This paper gives examples of different phenomena and explains how SAR signatures are manifested and how they can be interpreted with the help of other supplementary materials.

This published dataset focuses on the Eastern Mediterranean Sea, which is one of the marine oil pollution hotspots due to its heavy maritime traffic (Carpenter and Kostianoy, 2016). There are 1365 image patches with 3225 oil objects in *oil set* and 2990 image patches in *no-oil set*. Sect. 2 provides an overview of the dataset, including the spatial distribution of the data, the preprocessing steps applied to the SAR scenes, the procedures used to collect the dataset, and the sources of supplementary material used to understand the dark formations. Sect. 3 describes in detail how the oil slicks were annotated and the concept of the K-Means clustering algorithm used to classify the image patches.

The dataset and this paper can be used for different purposes. For researchers wishing to start in an area where there are no existing services for the recording of oil spills (i.e. no ground truth oil spills available), Sect. 4 can be used as a reference as it explains how oil slicks and different oceanic or atmospheric conditions contribute to SAR signatures. For studies that already have their own object detector for oil slicks, this dataset can be used as a separate dataset to test their model performance. Subsect. 5.1 provides an evaluation of the performance of an object detector used in an NRT automated oil spill detection system developed in Yang et al. (2024). In addition, the clustered *no-oil set* can provide an indication of how to improve the model. Furthermore, the dataset can be used as a training dataset by researchers who are just starting to apply object detection algorithms to oil spill detection applications. Although the dataset only covers oil spills in the Eastern Mediterranean Sea, a previous study showed that with such a locally focused oil object detector, only a small additional dataset is needed to extend the use of the detector to another region (Yang and Schnupfhagn, 2025). Additional technical information in the Subsect. 5.2 should be read before using the dataset. Section. 6 summarizes how this data descriptor and dataset can be used and add value to the community.

## 2 Materials

This section contains information about the dataset and has the following structure: Subsect. 2.1 lists general information about the SAR products and explains the corrections applied to them. Subsect. 2.2 gives an overview of the dataset and explains the procedures for preparing this dataset. Subsect. 2.3 shows a collection of different supplementary data that could help inspect dark formations. Note that all the time stamps shown in this paper and the dataset are in UTC. The map boundaries in figures were derived from Wessel and Smith (1996).

### 2.1 Sentinel-1 Data

Sentinel-1 SAR Level-1 Ground Range Detected (GRD) products were obtained from the Copernicus Open Access Hub, which provided data through the end of October 2023. The Copernicus Data Space Ecosystem operates as an improved and updated version of it and provides Earth observation data and services, including tools, graphical interfaces, and Application Programming Interfaces (APIs).

The dataset covers Sentinel-1A and Sentinel-1B products, which share the same orbit plane with a 180° orbital phasing difference; the repeat cycle was six days. Note that Sentinel-1B stopped delivering data since 23 December 2021. Sentinel-1 products from 2019 covering the Eastern Mediterranean Sea were acquired and preprocessed. The acquisitions from ascending and descending orbits in the area were taken at around 15:30–16:05 and 03:30–04:05, respectively. Those scenes are in Interferometric Wide Swath (IW) acquisition mode with a swath width of 250 km, and the incidence angle ranges between 29.1° and 46.0°. In this area, dual-polarization VV-VH products were provided. The data in cross-polarization mode (i.e., VH or HV) generally exhibit lower backscattering and, therefore, are influenced more by background and instrument noise compared to those in co-polarization mode (i.e., VV or HH) (Woodhouse, 2006). For this reason, only VV-polarized products were collected in the dataset.

A series of corrections, border noise removal, thermal noise removal, and calibration were applied to the SAR products. The continuous products were assembled and multilooked with a factor of 2. The final backscattering coefficient, $\sigma^0$, is given in decibels (dB). These preprocessing steps were done with the help of the Sentinel Application Platform (SNAP) Graph Processing Framework (GPF) (European Space Agency, 2020). Note that the data were preprocessed with SNAP version 8, and certain circular patterns were produced at regions with relatively low backscattering in a few images; Figure 1 shows such an example from 27 August 2019. Therefore, some image patches collected in the dataset also contain such artifacts (see Figure 2). As they should not be the bottleneck of the detection algorithm, these image patches are kept in the dataset.

Afterwards, the preprocessed data were normalized to 0–255, for training the object detector. The normalized image, $I_N$, was calculated by the sigmoid function:

$$I_N = \frac{255}{1 + e^{\frac{-(I-\beta)}{\alpha}}}, \tag{1}$$

where $I$ is the original image value. In this study, $\beta$ and $\alpha$ equal to the median and three times the standard deviation $(3 \cdot \sigma)$, respectively, of the original image values in the corresponding preprocessed data. There is no specific function that should be applied for image normalization. The parameters used to generate the dataset were determined by the authors through trial and error, as they provide good contrast between oil slicks and their surroundings according to human eyes. The object detectors shown in Subsect. 5.1 were trained using image patches for which a different image normalization method, provided by the Geospatial Data Abstraction Library (GDAL), was applied. Therefore, different image normalization methods may not be the key factor in the poor performance of the object detection algorithms. A previous study rescaled the image with maximum value equals to mean plus three times the standard deviation (Karathanassi et al., 2006).

## 2.2 Dataset

The published dataset utilized the annotation of oil objects in the framework of our previous study (Yang et al., 2024), where all preprocessed Sentinel-1 data covering the Southeastern Mediterranean Sea in 2019 were inspected jointly by two human interpreters. Only the dark formations agreed upon by both interpreters as oil slicks were given labels. Image patches including these labeled oil objects were generated and stored in the *oil set*. However, without ground truth oil spills available, the labeled

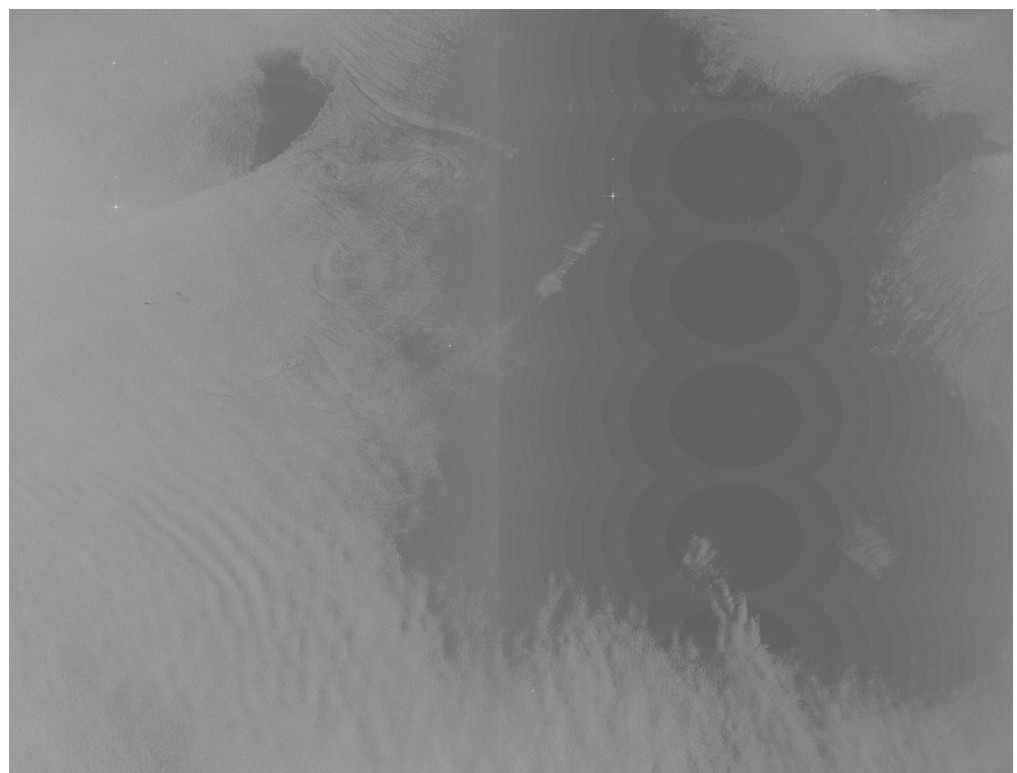

**Figure 1.** Preprocessed data with certain artifact produced after applying thermal noise removal in SNAP version 8. Note that the figure is a zoom-in of the entire preprocessed scene. The figure contains modified Copernicus Sentinel data [2019].

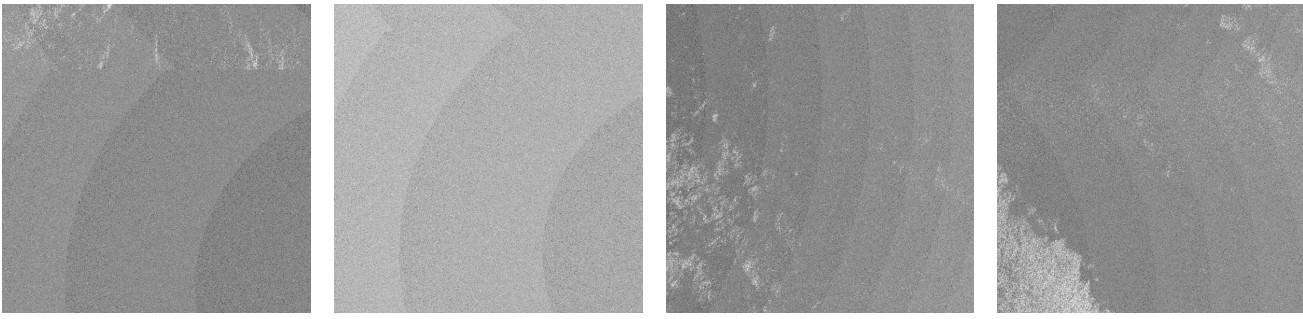

**Figure 2.** Image patches in the dataset with artifact as shown in Figure 1. The tags for these image patches (from left to right) for referring to their information in the provided data table are nw-0603-03-000100, nw-0609-03-000106, nw-0547-03-000044, and nw-0553-03-000050 (see also Subsect. 2.2 and Appendix A). The figure contains modified Copernicus Sentinel data [2019].

oil objects might not only belong to oil spills but also to other possible chemical spills. A successful oil spill detection system should be able to distinguish oil slicks from look-alikes; therefore, it is essential to take into account the image patches without

oil slicks in them. In addition to the *oil set*, the published dataset also includes a *no-oil set*, which provides image patches without oil slicks but with look-alikes or other remarkable SAR signatures present.

Sources of pollution and other phenomena could differ between coastal and offshore areas. Hence, depending on whether the image patch covers land areas or not, image patches in the *oil* and *no-oil sets* are further split into *coast* and *water* subsets. The structure of the different subsets can be understood as follows:

```
oil/
    coast/
    water/
no_oil/
    coast/
    water/
```

To efficiently collect image patches for the *no-oil set*, an object detector, custom-trained with oil objects but lacking images with look-alikes, was employed to target dark formations in the collected SAR scenes. These detections were compared to the locations of the inspected oil slicks, and those that have no intersection with the oil objects were regarded as look-alikes and stored in the *no-oil set*.

To include look-alikes from different sources equally, these image patches were categorized by the K-Means unsupervised clustering algorithm (see Subsect. 3.2 for algorithm explanations). The *water* and *coast* subsets were first separated with the help of a land mask, and then they were clustered into 12 and 5 subgroups, respectively. Afterwards, 2100 and 500 image patches were randomly chosen from the two subsets, respectively. As offshore regions are of more interest, more image patches from the *water* subset were kept in the dataset. Each cluster contains a similar number of selected image patches compared to the other cluster from the same subset. Figure 3 shows the numbers of image patches in each cluster; the red vertical lines show the number of image patches being randomly picked. It should be noted that one image patch might have dark formations from more than one source due to the complex manifestation of oceanographic phenomena on SAR imagery. Each subgroup should not be regarded as look-alikes from one specific source; the users should only consider the categories as a reference to help comprehend what kinds of SAR signatures are likely to be misinterpreted by their algorithms.

Sect. 4 provides explanations for SAR signatures from different ocean phenomena. The dataset includes image patches in JPG format, and the corresponding annotations are in Pascal VOC XML format (Everingham et al., 2010). Figure 4 shows the heatmap of image patches in the *oil* and *no-oil sets*. The oil spill inspection area is defined from 34.7° N to the south and 36° E to the east until the coastline and marked as a blue boundary in the figure. However, all the Sentinel-1 SAR products from 2019 covering the area in longitude from 30° E to 36° E and in latitude from 31° N to 34.7° N were automatically examined while collecting image patches for the *no-oil set*. As the oil inspection area is smaller than the distribution extent of the image patches in the *no-oil set*, these image patches were manually inspected, and the ones with oil slicks or unknown and unsure dark formations were removed from the dataset to avoid confusion. Table 1 gives the final statistics of the published dataset. Information about image patches in the dataset is recorded in a data table in Excel format; different subsets are recorded in

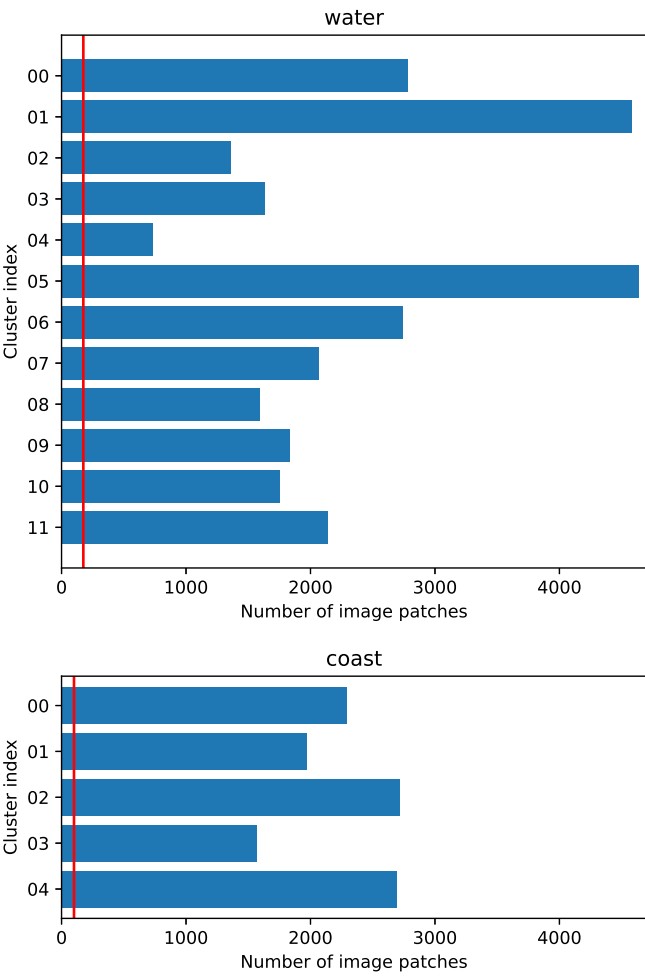

**Figure 3.** Numbers of image patches in different clusters by K-Mean algorithm. The vertical red lines indicate the number of image patches randomly selected from the subsets. These image patches were then manually inspected, and the number of image patches in the final published dataset from each subgroup is presented in Table 1.

separate sheets. Appendix A provides an extracted data table recording the information of image patches displayed in this article. For users who would like to preprocess SAR products themselves, Sentinel-1 product IDs and the corner coordinates of the image patches are provided. The data table includes the following fields relating to image patches:

- patch name,

• Sentinel-1 product start and stop date time,

- Sentinel-1 product ID,

- dimension of the image patch in pixels,

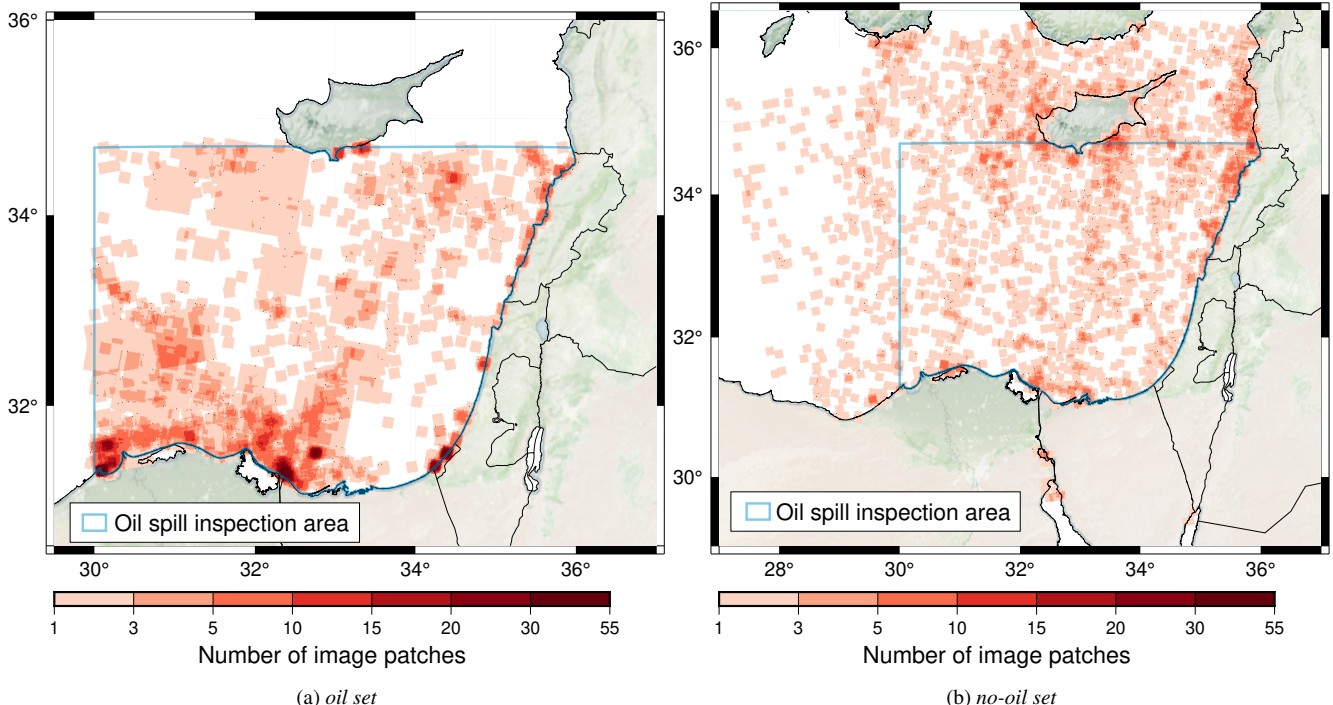

(a) *oil set*  (b) *no-oil set*

**Figure 4.** Heatmap of the image patches in the (a) *oil* and (b) *no-oil sets*. The base map was obtained from Stevens (2020), and the coastlines and borders were obtained from Wessel and Smith (1996).

- corner coordinates of the image patch in longitude and latitude (WGS84).

The patch name follows the naming convention:

`MM_YYYYMMDD_HHMMSS_HHMMSS_PP_i`

where `MM` refers to satellite mission, in this dataset, all the data are from Sentinel-1 mission, `S1`; `YYYYMMDD` shows the date of the product and `HHMMSS` shows the start and stop time of the product; `PP` indicates the polarization mode (e.g., VV); `i` is a series of numbers assigned when generated the image patches.

In the preprocessing step, continuous SAR products were assembled. Therefore, image patches might be located at the
overlapping areas of the two products; in this case, both Sentinel-1 product IDs would be listed in the data table. The dimension of the image patch is default to $640 \times 640$ px, which is the same as the model input size for our object detector; but to keep the original shapes of some large oil slicks, there are a few image patches that have dimensions greater than the default. In this case, the object detector would rescale these image patches, but oil slicks should still be detectable.

The tie points grids in Sentinel-1 products provide the geolocation information in WGS84 geocentric coordinate system. To
calculate the geolocation of the four corner points, upper left (ul), upper right (ur), bottom right (br), and bottom left (bl), of each image patch, its surrounding tie points were loaded, and the corner coordinates were estimated by the least squares fitting.

Table 1. Statistics of the *oil set* and *no-oil set*.

| Dataset | *oil set* | | *no-oil set* | | | |
|---|---|---|---|---|---|---|
| Subset | *water* | *coast* | *water* | | *coast* | |
| | | | 00 | 172 | 00 | 76 |
| | | | 01 | 171 | 01 | 84 |
| | | | 02 | 160 | 02 | 75 |
| | | | 03 | 162 | 03 | 44 |
| | | | 04 | 81 | 04 | 72 |
| | | | 05 | 166 | | |
| Number of image patches | 990 | 375 | 06 | 166 | | |
| | | | 07 | 171 | | |
| | | | 08 | 175 | | |
| | | | 09 | 168 | | |
| | | | 10 | 174 | | |
| | | | 11 | 173 | | |
| | | | | 1939 | | 351 |
| Number of oil objects | 2284 | 941 | N/A | | N/A | |

N/A: not applicable

The same procedure was used to obtain the object coordinates in *oil set*. For the *oil set*, additional fields for oil objects are provided:

- corner coordinates of the oil objects in longitude and latitude (WGS84),
- image coordinates of the oil objects referred to their corresponding image patch,
- the size (in pixels) of the bounding box annotating the oil object.

The data tables for *oil set* are sorted by objects; that is, if there are four oil objects in one image patch, there will be four separate rows for the four oil objects, and they should have the exactly same image patch information, such as patch name, Sentinel-1 product ID, and geolocation of the image patch. On the other hand, as there is no object in the *no-oil set*, each row refers to the information of its corresponding image patch. On top of all the fields explained earlier, each row in the table is referred to a unique tag with one image tag and an additional object tag or cluster tag.

The image tag is a two-letter subset tag followed by four-digit image serial number. The first letter of the subset tag is either o or *n*, referring to *oil set* and *no-oil set*, respectively; the second letter, *w* or *c*, stands for the subset *water* or *coast*. The image serial number is a sequence of numbers starting from 0001 for each subset, ordered by patch names.

In the *oil set*, each object gets an object tag with a two-digit object index and a six-digit object serial number. The object index numbers the objects in each image patch, counting from 01; the object serial number is a sequence of objects for all the objects in one subset, counting from 000001. As an example, ow-0795-01-001867 is a tag with an image tag ow-0795 and an object tag 01-001867, meaning that the object is the first object in the 795th image patch and the 1867th object in the *ow* subset.

To follow the same tag format, a cluster tag with two-digit cluster index and a six-digit cluster patch serial number is assigned to each image patch in the *no-oil set*. The cluster index refers to the class assigned by K-Means clustering methods, with 12 and 5 classes for *nw* and *nc* subsets, respectively; the index counts from 00. The cluster patch serial number is a sequence of image patches for all the image patches in one class, counting from 000001. Take nc-0308-04-000029 as an example, the image tag nc-0308 shows that this is the 308th image patch in the *nc* subset and the cluster tag tells that this is the 29th image patch in

the 04 class.

The patch name was assigned when the image patch was generated; it contains information about the date and time of the acquisition. Since the data publisher prefers a dataset without folders, the tags are used as filenames and the folders have been removed. However, users are encouraged to rename the filenames to the patch names if they prefer. However, it should be noted that image patches from the *oil* and *no-oil* sets were generated separately, so in some rare cases the image patches may have

the same names. Therefore, it is recommended to create a folder structure as explained at the beginning of this subsection.

### 2.3  Supplementary Data

Dark formations in SAR imagery can be caused by oil spills or look-alikes. Deliberate oil spills are mainly associated with human activities, such as offshore oil operations and oil transport. Some existing services, such as the Global Oil and Gas Extraction Tracker (GOGEC) and the European Marine Observation and Data Network (EMODnet), provide locations of oil,

gas and offshore installations. These services can help to identify areas with a high potential for platform spillage. On the other hand, look-alikes can be related to a variety of oceanic, atmospheric, and geological factors that modulate the roughness of the sea surface. Supplementary materials may be used to help in cross-comparison to better comprehend the sources of dark formations. Table 2 directs the reader to various services that the authors found valuable in explaining SAR signatures or that are used in the examples shown in Sect. 2.1.

Ocean information such as wind speed, waves, sea surface temperature (SST), and chlorophyll *a* (chl-*a*) concentration can be obtained from satellite, model, and in-situ data. The Ocean Virtual Laboratory (OVL) portal, funded by the European Space Agency (ESA), provides a quick and broad overview of ocean monitoring through its visualization tool. Additional information such as SAR roughness and bathymetry is also available. The other service, the Copernicus Marine Data Store (MDS) from the E.U. Copernicus Marine Service Information (CMEMS; acronym derived from its former name, Copernicus

Marine Environment Monitoring Service), is an ocean data catalog with products at global and regional scales. Similar to the OVL portal, MDS provides MyOcean Viewer, which allows users to add different products to the map viewer. Moreover, the dataset can be easily retrieved with commands using the Copernicus Marine Toolbox.

**Table 2.** Platforms for supplementary data.

| Platform | url | Reference |
|---|---|---|
| Ocean Virtual Laboratory (OVL) portal | https://ovl.oceandatalab.com/ | |
| Copernicus Marine Data Store (MDS) | https://data.marine.copernicus.eu/ | |
| • Mediterranean Sea - High Resolution and Ultra High Resolution L3S Sea Surface Temperature | https://doi.org/10.48670/moi-00171 | CMEMS (a) |
| • Mediterranean Sea, Bio-Geo-Chemical, L3, daily Satellite Observations (1997–ongoing) | https://doi.org/10.48670/moi-00299 | CMEMS (b) |
| • Global Ocean Hourly Sea Surface Wind and Stress from Scatterometer and Model | https://doi.org/10.48670/moi-00305 | CMEMS (c) |
| Copernicus Climate Data Store (CDS) | https://cds.climate.copernicus.eu/ | |
| Israel Meteorological Service | https://ims.gov.il/en | Ministry of Transport and Road Safety, Israel |
| General Bathymetric Chart of the Oceans | https://www.gebco.net/ | GEBCO (2023) |

Another Copernicus service, the Climate Data Store (CDS), provides climate data such as global precipitation data; this service is implemented by the European Centre for Medium-Range Weather Forecasts (ECMWF). However, the rainfall data used in this article were obtained from the coastal weather stations obtained from the Israel Meteorological Service, provided by the Ministry of Transport and Road Safety, Israel. The service provides daily rainfall and rainfall automated recorded every 10 minutes. Note that these systems tend to underestimate the rainfall in major events according to the information provided by the service.

The General Bathymetric Chart of the Oceans (GEBCO) has released several global bathymetric and topography grids with a spatial resolution of 15 arc seconds since 2003. This model is especially useful for providing context to SAR data in areas with varying bathymetry, which can affect the interpretation of ocean surface roughness in satellite imagery.

Sect. 4 explains different SAR signatures and provides examples of explanations on different dark formations with the help of the supplementary data. A summary list for those examples along with the supplementary data is provided in Subsect. 4.9 (see Table 3). Note that not all the data in the dataset were confirmed with the supplementary data but with the experience of the human inspectors and their understanding of the study area.

## 3 Methods

### 3.1 Manual Inspection

As stated in Subsect. 2.2, the annotations were initially done in the framework of our previous study (Yang et al., 2024) with the help of an open source image annotation tool, LabelImg (Tzutalin, 2015). Those image patches were geolocated, as in this case, referring to other supplementary materials and understanding the location of the spills is easier. However, the image patches and the oil objects in the published dataset were provided in the range and azimuth direction of the corresponding SAR products.

Continuous oil spillage from one source can be like multiple oil slicks after the physical processes. Therefore, definitions of those oil objects might be tricky; one can interpret them as one oil object or several oil objects. In addition, the extent of an oil slick might not be easy to define; for example, the evaporation of oil could make it look like it is fading away in SAR scenes. It shall be noted that different definitions of the extent of one oil object can play a vital role in model performance calculations. For example, if there is one oil object in the dataset, but the object detector considers it to be two nearby objects, they may be considered false positives if they do not pass the threshold used to define true positives (see Subsect. 5.1 for performance evaluation). Figure 5 provides some image patches inside the published dataset with oil object labels to illustrate how the authors annotated the oil slicks. The tags of these image patches are provided as captions, and the explanations for tags are shown in Subsect. 2.2; readers can find an extracted data table in Appendix A.

### 3.2 K-Means Clustering

K-Means (Lloyd, 1982; MacQueen, 1967) is an unsupervised algorithm to partition a set of observations into a specific number, $k$, of clusters. The concept is to obtain $k$ clusters by satisfying that the sum of the distance between each observation and the mean vector (centroid) of its corresponding cluster is minimum; the Euclidean distance is a common way of calculating such distance. To achieve this idea, the algorithm first randomly partitions observations into $k$ sets and calculates their centroids. Based on these centroids, each observation is assigned to its nearest cluster. Afterwards, the algorithm iteratively updates the centroid and assign observations of each cluster to a new cluster until convergence is achieved.

In image clustering, each image is represented as a feature vector in the vector space that contains the features of the image, such as texture and shape. To extract features of the image patches in the *no-oil set*, the InceptionV3 model (Szegedy et al., 2015) pre-trained with a large visual database, ImageNet (Deng et al., 2009), was used. K-Means clustering was then applied to categorize the image patches with the help of the Scikit-learn K-Means module. Figure 6 shows examples of image patches from each cluster (see Appendix A for the patch information).

The purpose of using the K-Means clustering method is to provide a *no-oil set* in which the signatures of different phenomena appear more balanced. In other words, the clustering method categorized the dataset, and then the same number of image patches from each class were randomly selected. However, for better understanding the performance of the object detector on different kinds of SAR signatures, it would be ideal if all image patches within each class have similar sources of phenomena

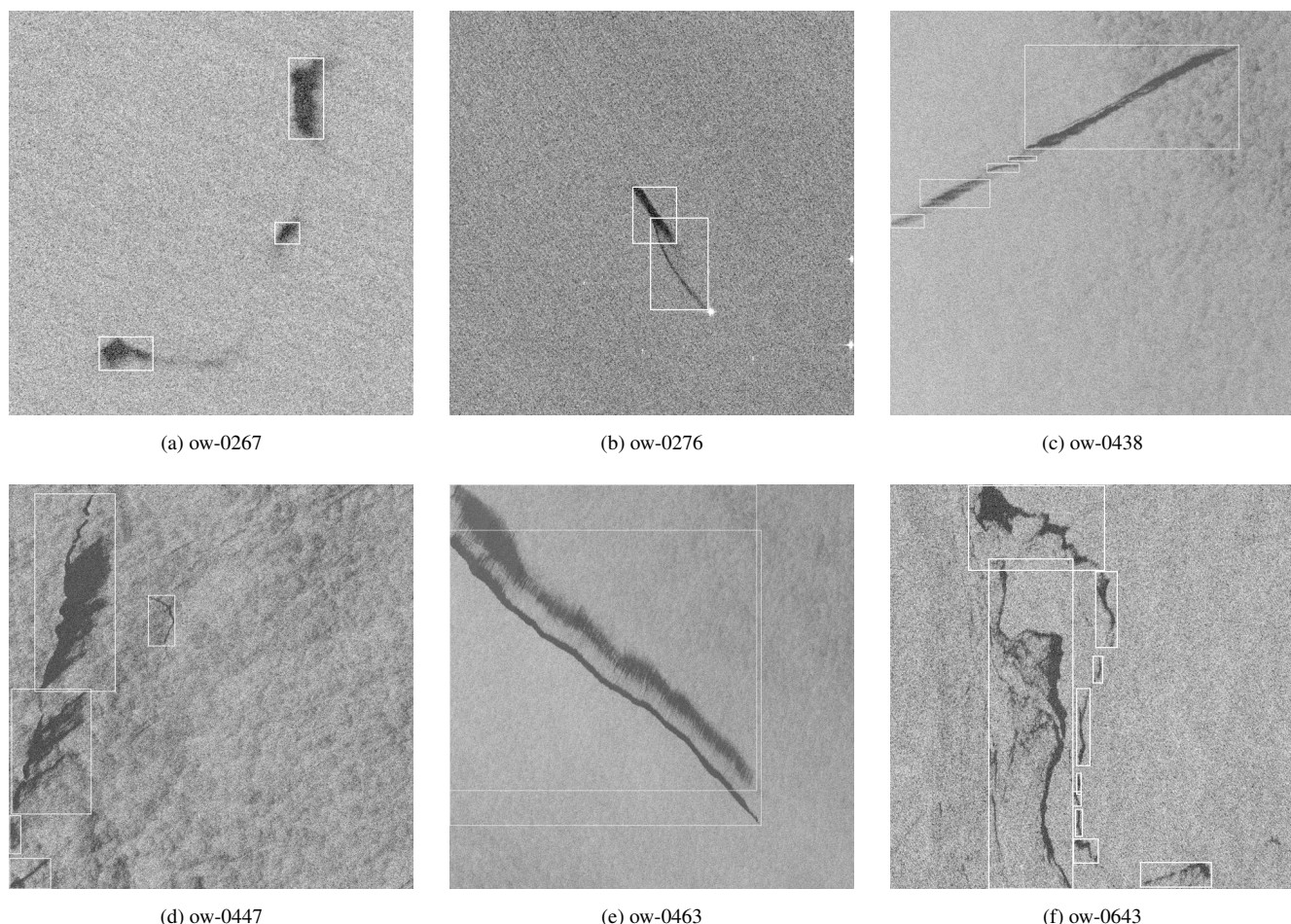

| (a) ow-0267 | (b) ow-0276 | (c) ow-0438 |
| (d) ow-0447 | (e) ow-0463 | (f) ow-0643 |

**Figure 5.** Image patches from the published dataset along with the labels of the oil objects marked with white bounding boxes. Their corresponding image tags (explained in Subsect. 2.2) are also provided as captions, the readers can find their further information recorded in the data table (see Appendix A). The figure contains modified Copernicus Sentinel data [2019].

that could be explained by humans. Therefore, human interpretation was considered when selecting the number of classes, which was adjusted by reviewing image patches in different classes.

Sea states in coastal areas are often complicated by factors such as bathymetry and interaction with land. However, open water typically experiences larger-scale dynamics that can manifest a variety of different SAR signatures. Therefore, the offshore subset is expected to have a greater variety of sources of SAR signatures. Thus, the K-means clustering method was given 12 and 5 classes for the *nw* and *nc* subsets, respectively.

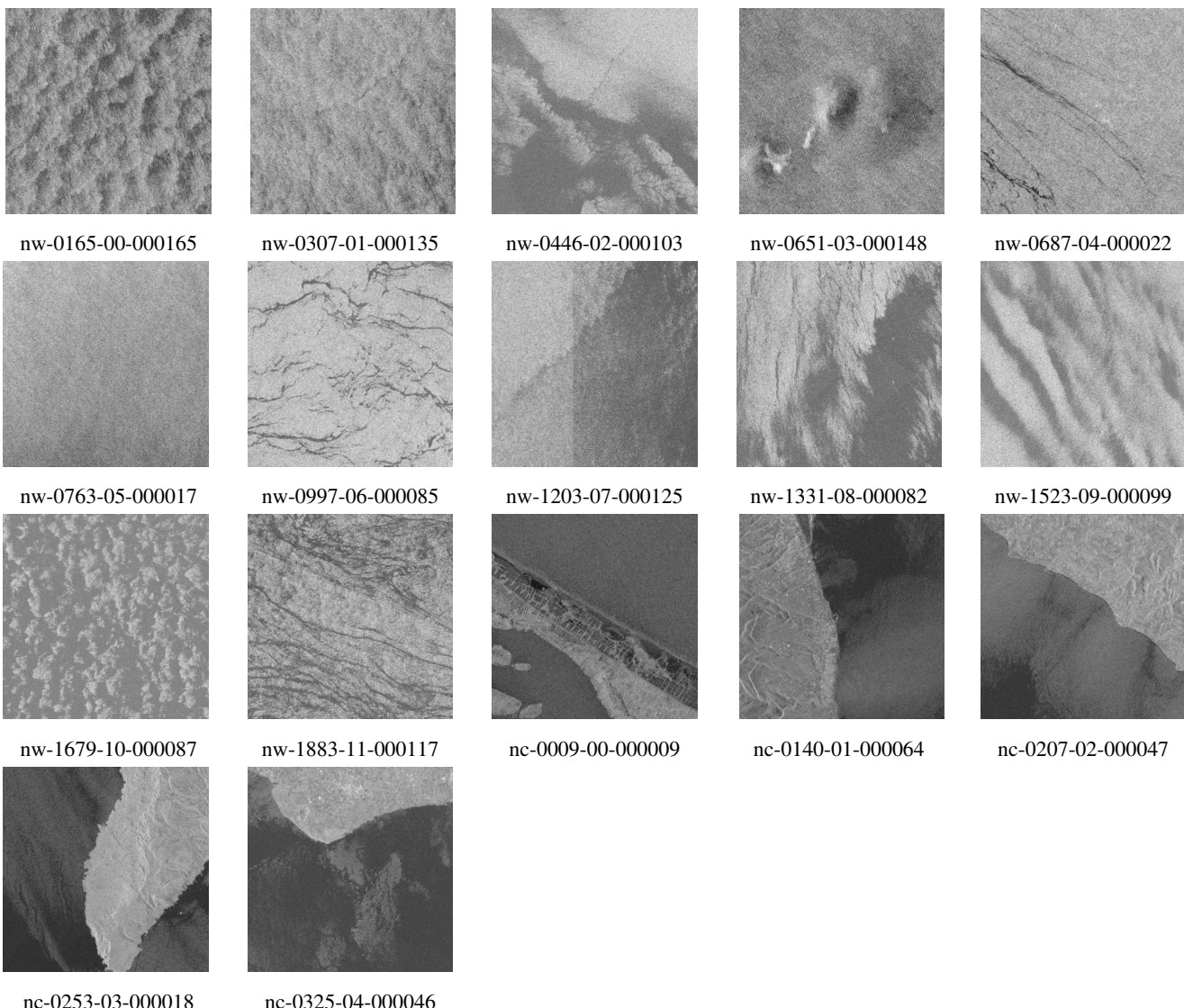

nw-0165-00-000165     nw-0307-01-000135     nw-0446-02-000103     nw-0651-03-000148     nw-0687-04-000022

nw-0763-05-000017     nw-0997-06-000085     nw-1203-07-000125     nw-1331-08-000082     nw-1523-09-000099

nw-1679-10-000087     nw-1883-11-000117     nc-0009-00-000009     nc-0140-01-000064     nc-0207-02-000047

nc-0253-03-000018     nc-0325-04-000046

**Figure 6.** Image patches from different clusters in the *no-oil set*. Their corresponding tags (explained in Subsect. 2.2) are also provided as captions, the readers can find their further information recorded in the data table (see Appendix A). The figure contains modified Copernicus Sentinel data [2019].

## 4   SAR Signatures

The radar transmits microwave pulses and some of them are reflected back to the radar, the normalized power of the received signals over the actual ground area is known as radar backscatter, $\sigma^0$. The radar backscatter depends on radar system characteristics (e.g., polarization, wavelength, and radar geometry) and the properties of the target (e.g., shape, dielectric constant, and roughness) (Woodhouse, 2006).

For ocean applications, sea surface roughness is generally regarded as a key factor; variations in surface roughness are closely related to wind speed and direction (Robinson, 2004; Woodhouse, 2006). Winds form friction between air and water and cause small capillary waves in millimeter-to-centimeter scales. These wind-induced capillary waves are usually regular over a large area and act as resonant Bragg scatterers, which can interfere constructively if the Bragg condition is satisfied as defined:

$$2d \cdot \sin\theta = n\lambda, \tag{2}$$

where $d$ is the spacing of the scatterers, $\theta$ is the incidence angle, $\lambda$ is the radar wavelength, and $n$ is an integer. The spacing of the scatters can be regarded as ocean wavelength. The incidence angle of Sentinel-1 IW mode ranges between 29.1° and 46.0°; therefore, according to the equation, the resonant ocean wavelength is at a similar scale as radar wavelength. In addition, the waves should travel along the range direction (either parallel or anti-parallel) to obtain the strongest resonance. This constructive interference is also known as resonant Bragg scattering or the coherence scattering mechanism. On the other hand, if the condition is not fulfilled, destructive interference occurs, and the scattered power is reduced.

The ocean surface contains small-scale capillary waves, gravity waves in meter scales, swell, and large-scale currents; therefore, the ocean surface is considered a complicated summation of a wave spectrum of different wavelengths. However, the radar returns come from these short capillary waves, which have wavelengths similar to the radar wavelengths, rather than the longer waves; hence the Bragg mechanism is often used to interpret radar backscatter at the ocean surface (Robinson, 2004).

The following subsections illustrate and explain SAR signatures due to oil slicks, different oceanic or atmospheric phenomena, or human-related activities. Some supporting materials listed in Table 2 are used to comprehend the phenomena and to better interpret the SAR signatures. If users need more detailed information about the supporting data, they should check the descriptions in Sect. 2.3 or refer to the websites in Table 2. Note that these examples are not provided in the published dataset. The published dataset only includes image patches from 2019; however, the examples are not limited to the time interval. The selections are mainly based on the accessibility of the supplementary materials and are to avoid SAR signatures due to multiple phenomena, which make it challenging to interpret. These examples are not cropped into image patches as most phenomena influence an area larger than the size of an image patch. In addition, instead of displaying them in SAR geometry in range and azimuth, they were projected to the World Geodetic System 1984 (WGS84), making it easier to compare them with different supplementary data. All the example scenes are listed at the end of this section (Subsect. 4.9), so that users can download SAR scenes themselves from the Copernicus Data Space Ecosystem.

## 4.1 Oil slicks

The presence of oil slicks decreases the surface tension of the water and lowers wind friction; therefore, short gravity and capillary waves are dampened, which reduces the radar backscatter and results in dark formation in SAR imagery. Spaceborne SAR sensors can observe a large area on a regular basis; however, to investigate the quantity, type, and thickness of the oil, it usually requires multiple sensors, such as infrared, ultraviolet, microwave radiometer, and laser fluorosensors (Ferraro et al.,

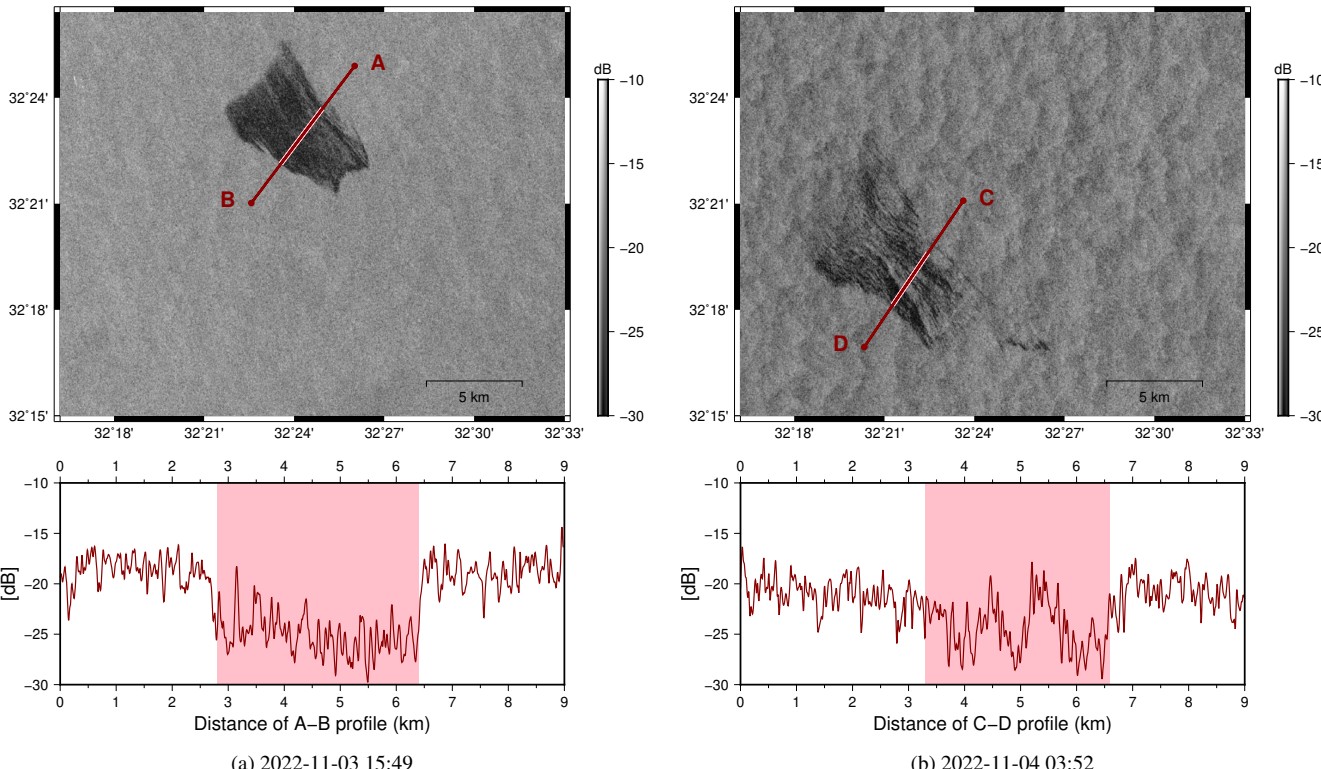

**Figure 7.** Examples of an oil slick appearing in SAR acquisitions around 12 hours apart, showing its changes overtime. The profiles illustrate the radar backscatter along the red line, and the shadowed area highlights approximately where the oil-induced dark formation is located. The figure contains modified Copernicus Sentinel data [2022].

2009; Fingas and Brown, 2017). This dataset was inspected without data from other sensors available; therefore, the distinction between oil spills and other possible chemical spills is not clearly defined.

Figure 7 shows SAR scenes covering an oil slick in two separate acquisitions taken at 15:49 on 3 November 2022 and at 03:52 on 4 November 2022. Separate charts show the profiles of radar backscatter sampled along the red lines, the shadowed areas mark the approximate locations of oil-induced dark formations. Wind speeds at 16:00 on 3 November 2022 and 04:00 on 4 November 2022 were acquired from CMEMS (c); the corresponding average wind speeds of the shown area were $3.94\,\mathrm{m\,s^{-1}}$ and $3.90\,\mathrm{m\,s^{-1}}$, both blowing in the southeastward direction. The first acquisition shows an oil slick with apparent differences in backscattering compared to its surrounding areas. Though the wind conditions seemed similar, the two acquisitions had different heading and look angles; therefore, the wind-induced signatures differed and resulted in lower overall backscattering in the second acquisition. This example illustrates the dynamicity of an oil spill over time in SAR data and how oil properties, ambient wind speed and direction relative to the SAR azimuth make oil slicks appear differently in SAR data.

Reports or records are usually not provided for smaller deliberate oil discharges, meaning that there is no ground truth; thus, inspecting oil slicks should carefully consider the possibilities of dark formations from other phenomena. The following subsections introduce those *look-alikes* and provide supplementary materials to better understand the signatures.

## 4.2 Wind

As explained at the beginning of this section, wind is closely related to the sea surface roughness, which is a key element for SAR signatures. Under low winds, the sea surface is smooth and calm, and the backscattering will be close to the SAR noise floor; therefore, the modulation of backscattering from oil slicks can not be revealed. On the other hand, if the sea surface is too rough under high winds, oil slicks are also not possible to be indicated. Previous studies suggested optimal wind speed ranges between $2$–$3\,\mathrm{m\,s^{-1}}$ and $7$–$12\,\mathrm{m\,s^{-1}}$ for oil slick detection using SAR (Gade et al., 2000; Robinson, 2004; Brekke and Solberg, 2005). As the visibility of oil slicks depends on not only SAR sensors but also the age and type of the oil, the upper and lower limits differ in different studies; this range should be considered a hint. Oil slicks can still be observed outside this range in some circumstances.

Some atmospheric phenomena can influence ocean surface roughness and result in certain SAR signatures; examples of these phenomena are the atmospheric front, phenomena related to geographic features, unstable atmospheric boundary layer (ABL), and atmospheric gravity waves (AGWs, also known as atmospheric internal waves) (Robinson, 2004). Since AGWs form similar patterns as oceanic internal waves (OIWs), they are both explained in Subsect. 4.3.

Sudden changes in wind speed and direction can create an atmospheric front that separates two air masses and is shown as a boundary between weaker and stronger radar backscatter. Similar signatures of the fronts can also be seen where winds blow from land to sea through coastal terrain. The land usually cools down during the night, but the temperature over the adjacent sea may remain, in which case the air pressure over the land would be higher than that over the sea, resulting in the wind blowing offshore. This cooler and denser air from the land rolls out to the sea and pushes the warm air over the sea upwards, creating a cold land breeze front (Robinson, 2004). Land breeze fronts usually create near-shore bands of modulated surface roughness parallel to the coast, as shown in Figure 8. Similarly, during the night, the cool air mass from the mountains could flow down the valley driven by the density flow; this is known as the katabatic wind. If the mountains are located in coastal areas, the winds would blow toward the sea and spread out in a fan shape, as shown in Figure 9 (a). Figure 9 (b) indicates the topography with mountains and valleys close to the shore in Lebanon; the terrain and bathymetry model is obtained from GEBCO (2023). These katabatic wind-induced signatures tend to reoccur at roughly the same places since they are related to the topography (Robinson, 2004). Note that the bright pixels are especially distinct at the bottom part of Figure 9 (a) was due to radio frequency interference (RFI), which will be explained in Subsect. 4.8.

Figures 8 and 9 (a) were both taken at about 03:44 UTC, which should be 06:44 local time (UTC+3, considering daylight saving time); shortly after sunrise at about 06:03 and 06:29 local time, respectively. Land breeze and katabatic winds usually occur during the night, when the land has cooled down a lot, and before the air warms up during the day. Based on our experience inspecting SAR images in this area, the katabitic wind fronts are commonly observed in SAR images in the de-

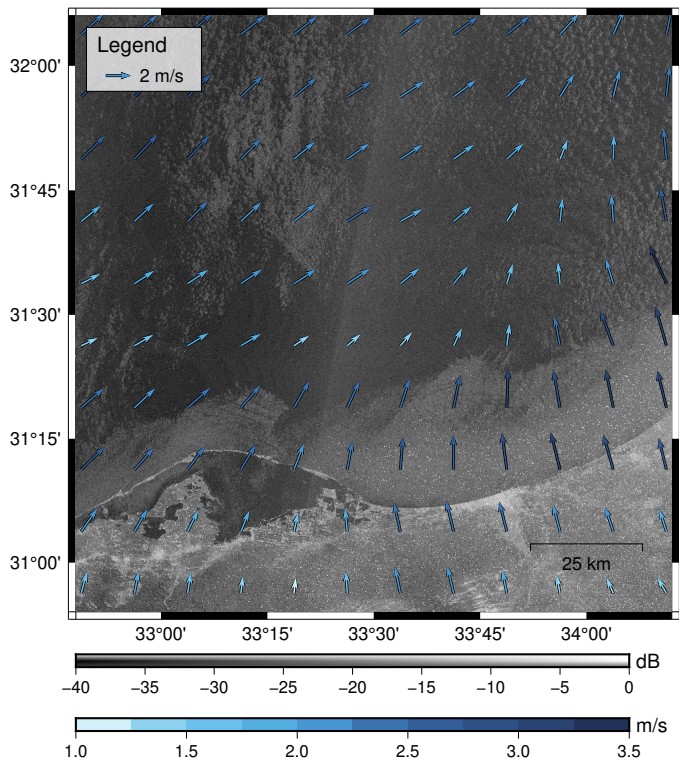

**Figure 8.** An example of land breeze fronts in the SAR scenes taken at 03:44 on 2 August 2023 near Egypt, plotted with the simulated wind speed at 04:00 obtain from CMEMS (c). The figure contains modified Copernicus Sentinel data [2023]. The plotted coastline was obtained from Wessel and Smith (1996).

scending orbit (taken at around 03:45 UTC) covering the coastal areas of Lebanon and Israel. In addition, wind shelters and wind shadows can also appear as low radar backscatter along the coast or off an island.

     Dynamics in the ABL driven by surface wind can increase surface roughness and lead to phenomena seen in SAR imagery. When the air is heated by a warmer sea surface, it expands and becomes less dense than the air above it. The resulting ABL instability drives convective cells in which warm and humid air rises in updrafts and is replaced by descending cooler air. The

downflow of the cold air induces a radial outflow; coupled with the wind flow, they lead to the fluctuations of surface roughness and form cellular patterns in SAR images (Robinson, 2004); such patterns can be observed in Figures 8 and 9 (a).

     Thermodynamic instability in ABL, mainly determined by temperature and humidity, can lead to wind streaks forming streak-like patterns in SAR (Zhou et al., 2025). These wind streaks enable the estimation of wind direction by using SAR (Lehner et al., 1998). Similar streak patterns may also result from the intersection of wind-driven surface currents and

Stoke's drift of surface waves. This intersection can lead to Langmuir circulation, producing helical roll vortices that are approximately parallel to the wind direction (Langmuir, 1938; Etling and Brown, 1993). These helical rolls appear alternatively as the right and left helices, resulting in upward and downward flows between the rolls, which lead to higher and lower surface

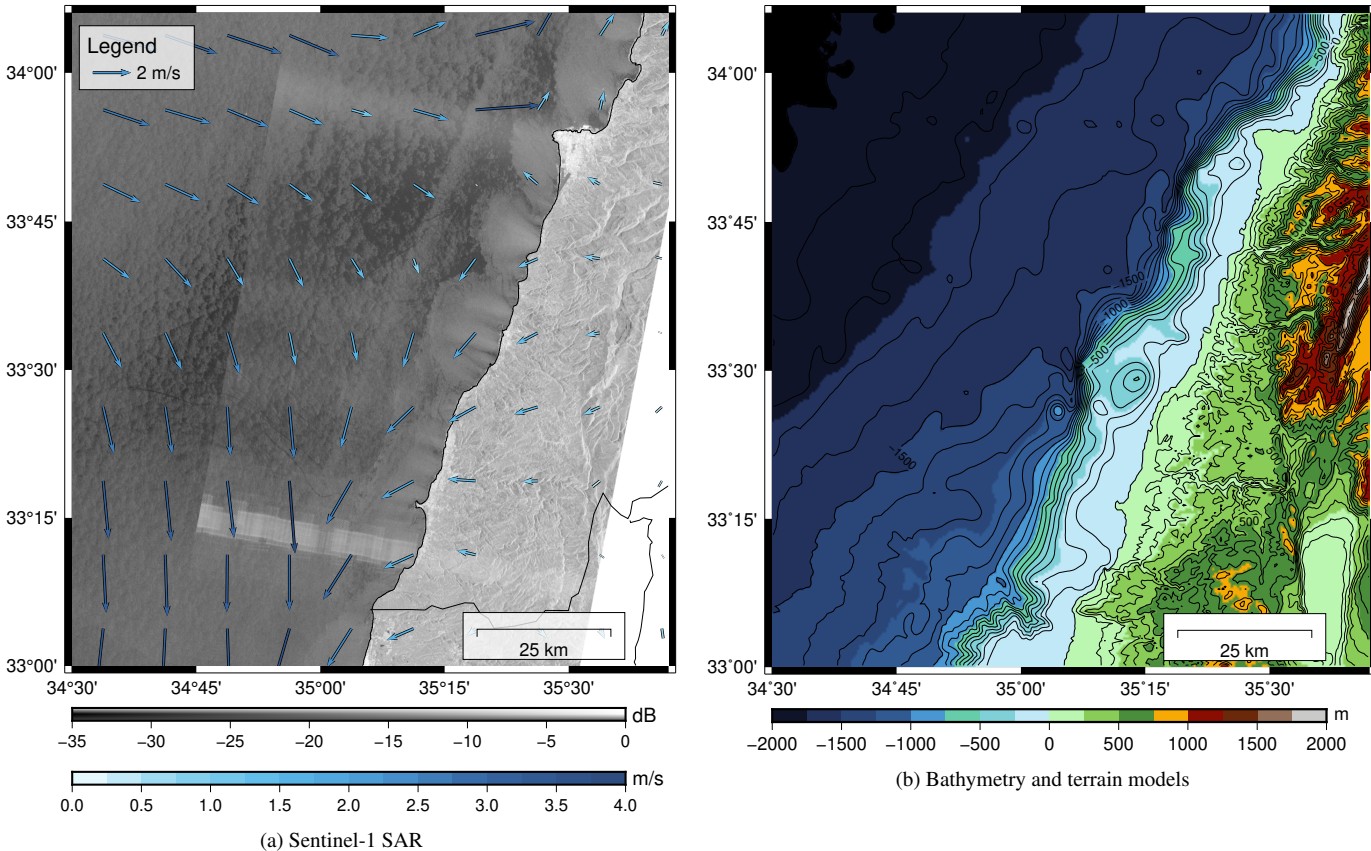

**Figure 9.** An example of katabatic wind-induced signatures in the SAR scenes taken at 03:43 on 24 September 2022 near Lebanon, plotted with the simulated wind speed at 04:00 obtain from CMEMS (c). The left figure contains modified Copernicus Sentinel data [2022]. The coastline and borders were obtained from Wessel and Smith (1996).

roughness and are shown as dark and bright streaks in SAR (Langmuir, 1938; Robinson, 2004). Figure 10 shows an example of such streak patterns alongside the wind speed and direction.

### 4.3 Internal Waves in the Ocean and Atmosphere

Internal waves can occur in any stratified medium, such as fluids with varying density. In the ocean and atmosphere, two restoring forces act on internal waves: gravity and the Coriolis force. Thus, perturbations of the vertical density gradient will generate internal waves with frequencies between the Brunt-Väisälä frequency and the Coriolis parameter.

The primary drivers of internal waves in the ocean are tides, closely followed by the wind. The particle motion of the oceanic internal waves (OIWs) produces surface convergence and divergence that modulate the short gravity and capillary waves, resulting in amplification and attenuation of these waves, respectively. As a result, the alternating convergence and

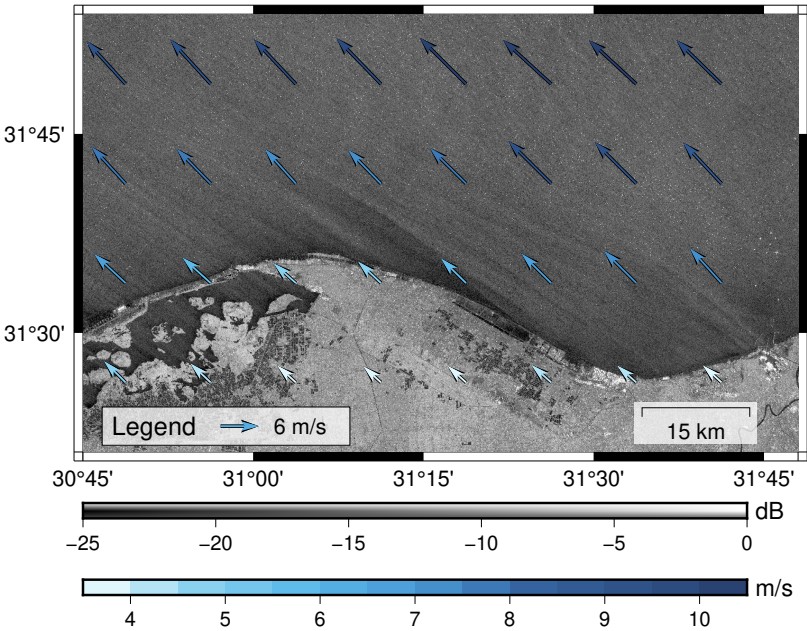

**Figure 10.** An example of streak patterns in the SAR scene taken at 03:52 on 27 January 2023 near Egypt, plotted with the simulated wind speed at 04:00 obtain from CMEMS (c). The figure contains modified Copernicus Sentinel data [2023].

divergence zones on the sea surface lead to patterns of bright and dark strips in SAR imagery (Alpers et al., 2008; Robinson, 2010; Alpers and Huang, 2011). Similar to the ocean, disturbances in atmospheric stratification also generate internal waves, often referred to as atmospheric gravity waves (AGWs).

When for example external forces push the air upward, the air gets cooler, and the water vapor saturation point is likely to be reached, which leads to the formation of clouds. On the other hand, when the air gets warmer, water evaporates, resulting in a clear sky. Therefore, when the moisture content of the air is sufficient and the amplitude of the AGW is large enough, cloud streets and clear skies are expected to appear over the crests and troughs of the AGW, respectively. The variations of wind stress at the sea surface disturb the small surface roughness and result in dark and bright strips in SAR scenes, similar to
OIW (Alpers and Huang, 2011).

Internal waves play a crucial role in energy transport within the ocean and atmosphere. The interactions of internal waves with itself, topography and other ocean or atmosphere dynamics are highly complex and not yet fully understood. Through the SAR images and the identification and clustering of possible OIW and AGW signatures by the algorithm, there is now a high number of images available, providing a better spatial and temporal resolution of the internal wave field and allowing
for a deeper understanding of these processes. However, it can be challenging to distinguish between OIWs and AGWs in general and within the presented dataset. Nevertheless, the shape and structure of the wave patterns provide hints for their differentiation (Alpers and Huang, 2011), as summarized in the following. In general, OIWs are observed in low-wind areas since otherwise their SAR signatures are too weak to be detected, especially in conditions with wind speeds greater than

$10 \, \mathrm{m \, s^{-1}}$; contrastingly, AGWs can be observed at all wind speeds. In addition, OIWs mostly appear near an upwelling area or
395 at locations where rough topography, shallow underwater ridges, sea mounts, or steep shelves are present. On the other hand, AGWs usually appear in areas where wind interacts with mountain ranges, different air masses collide, strong wind shear occurs, or convective activities are associated with cold fronts (e.g., thunderstorms).

A previous study presented OIWs propagating from the edge of the continental shelf and supported their explanation with temperature, salinity, and density profiles from a CTD probe (Liu et al., 1998). Figure 11 shows the SAR scenes taken at
400 15:41 on 28 March 2024 at the coast of Lebanon, plotted with the contour of bathymetry and terrain model obtained from GEBCO (2023). The bathymetry profile along the red line is shown in a separate chart. Dark and bright strips can be seen east of the continental slope (between points B and A). The strips appear to be parallel to the both the coastline and the continental slope. Therefore, the stripes me be due to oceanic internal waves generated in the interaction between currents and the slope, or due atmosperic internal waves generated as lee waves from the mountain topography. Without additional data, such as vertical
profiles of ocean and atmosphere, it is difficult to determine the sources of internal waves in this example.

Figure 12 shows the SAR scenes taken at 03:52 on 27 May 2023 and a separate profile of radar backscattering along the red line. On top of the SAR scene, the hourly sea surface wind at 04:00 on 24 May 2023 obtained from the scatterometer and model (CMEMS, c) is plotted; the wind speed in this area was around $2.87 \, \mathrm{m \, s^{-1}}$ to $10.18 \, \mathrm{m \, s^{-1}}$, and the average wind speed was $7.97 \, \mathrm{m \, s^{-1}}$. The higher the wind speed, the less likely it is to be OIWs; therefore, the strip patterns are more likely to come
from AGWs. Based on the consecutive scenes taken before (i.e., northern to) the shown scene, the wave patterns cover large ocean areas, which are commonly observed in SAR scenes with AGWs (Li, 2004). In addition, streaks from AGWs are usually approximately perpendicular to the direction of the wind (Robinson, 2004; Li, 2004) and are expected to show up as narrow dark bands followed by broad bright bands alternatively (Alpers and Huang, 2011).

## 4.4 Areas of mixing and vertical advection in the ocean

In the ocean, there are a variety of processes that can lead to mixing or vertical advection, especially near shallow topography or the coast. In the deeper layers of the ocean, more nutrients and colder water can be found compared to the upper layers. This means that mixing or vertical advection typically results in a colder sea surface and a vertical nutrient flux, which in turn promotes chlorophyll growth. Mixing can be caused, for example, by the mentioned OIWs, which, like surface waves, can break, leading to mixing. Vertical advection typically occurs due to strong, steady winds, which create a force on the surface
layer of the sea, causing it to move in the wind direction. The Coriolis force then deflects this motion to the right in the northern hemisphere (left in the southern hemisphere), a process called Ekman transport. This results in a 90° shift in the surface layer's movement. This divergence of Ekman transport brings cooler, nutrient-rich water from deeper layers to the surface, known as upwelling, often seen along coastlines where winds blow parallel to the shore (Robinson, 2010; Knauss and Garfield, 2016).

Previous studies have shown areas of enhanced mixing and/or vertical advection in SAR imagery with various causes, such as
coastal parallel winds or mixing within cyclonic eddies (Clemente-Colon and Yan, 1999; Alpers and Zeng, 2021). In upwelling regions, low backscatter in SAR imagery can be observed due to increased stability of the ABL (due to reduced wind stress from the lower SST), increased surface water viscosity (which enhances the dampening effect), and the presence of biogenic

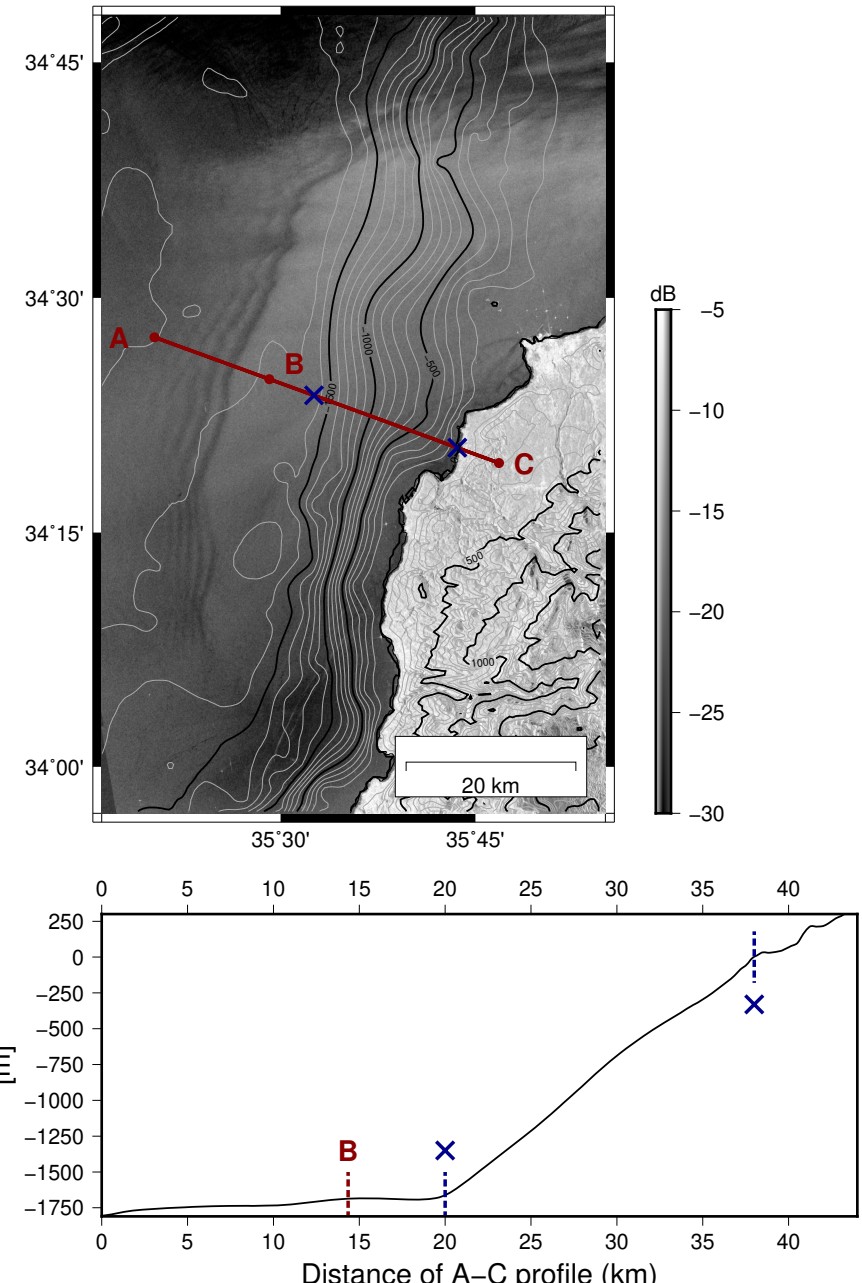

**Figure 11.** An example of internal waves in the SAR scene taken at 15:40 on 28 March 2024 near Lebanon, plotted with the contour of bathymetry obtain from GEBCO (2023). A separate chart illustrates the bathymetry profile along points A and C, sampled every 50 m. The figure contains modified Copernicus Sentinel data [2024].

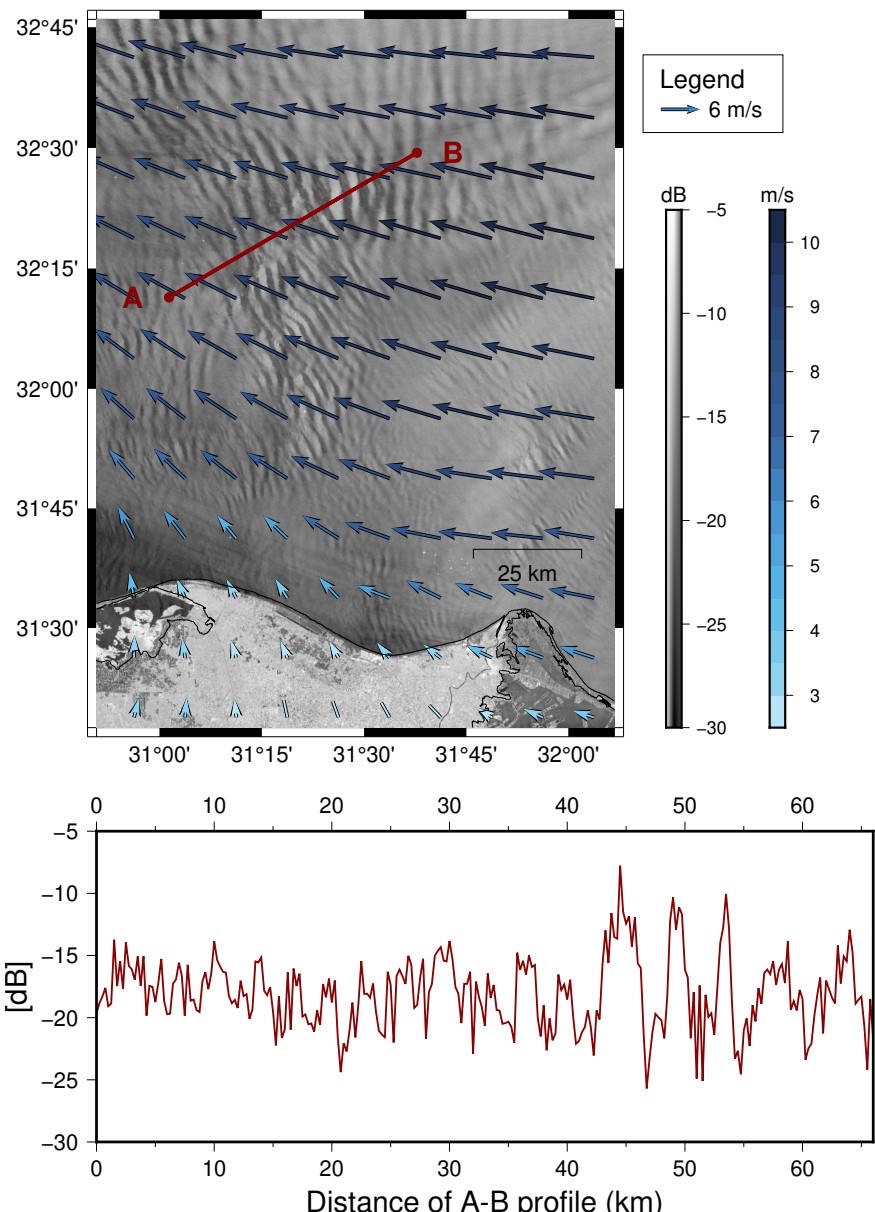

**Figure 12.** An example of AGWs in the SAR scene taken at 03:51 on 27 May 2023 near Egypt, plotted with the sea surface wind velocity obtain from MDS (CMEMS, c). A separate chart shows the profile of radar backscatter along the red line, sampled every 250 m. The figure contains modified Copernicus Sentinel data [2023]. The coastline was obtained from Wessel and Smith (1996).

surface films (see Subsect.4.6) (Clemente-Colon and Yan, 1999). In addition to these mechanisms, the surface divergence and convergence may also play a role (Liu et al., 2016). Previous studies provided a comprehensive explanation of the areas of enhanced mixing or vertical advection in the Mediterranean Sea, for example Bakun and Agostini (2001). In the region of our

dataset, the following areas (mostly wind-induced upwelling zones), can be found near the coast: The coastal divergent zone on the south side of Cyprus tends to induce upwelling, especially in summer. On the other hand, the north side of Egypt is dominated by coastal convergence with downwelling, where surface water is brought downward with the flow throughout the year but is most intense in winter. Along the eastern boundary of the Mediterranean Sea, the coasts of Syria, Lebanon, and Israel, the wind generally blows eastward toward the coast. However, the slight right turnings of the wind induce some areas of upwelling.

Figure 13 (a) shows SAR scenes covering Cyprus at 03:51 on 12 May 2019 and illustrates a possible coastal upwelling area at the southern coast of Cyprus. As suggested in Alpers and Zeng (2021), SST and chl-*a* data can be used to help indicate upwelling areas; Figures 13 (b) and 13 (c) present the corresponding data at 00:00 on 12 May 2019 obtained from the SST MDS (CMEMS, a; Buongiorno Nardelli et al., 2013) and chl-*a* MDS (CMEMS, b; Berthon and Zibordi, 2004; Volpe et al., 2019). In most areas, the patterns of both data were similar to each other. However, the southwestern coast of Cyprus had a lower chl-*a* concentration than the southeastern coast, though the southwestern and southeastern coasts had similar SST. On the other hand, SAR scenes showed low radar backscatter at both the southwestern and southeastern coasts. Wind speeds were about 2–3 m s$^{-3}$ in the coastal areas according to the hourly sea surface wind velocity from scatterometer and model at 04:00 on 12 May 2019 obtained from MDS (CMEMS, c). Therefore, the low radar backscatter could result from low wind speeds coupled with surface divergence due to upwelling. Additionally, SAR scenes indicate possible surface films that have accumulated as linear features to the west of the dashed red line in the figure. This aligns well with the patterns of SST and chl-*a* concentration.

## 4.5  Meso- and Submesoscale Eddies

Mesoscale eddies (the prefix "meso" means "intermediate") describe features with radii of about 10 to 200 km and a lifetime of a few days to one year or even longer (Chelton et al., 2007). Submesoscale is defined as slightly smaller than the mesoscale, with horizontal scales of 100 m to 10 km (or less than the first baroclinic mode Rossby radius of deformation (Rd)), vertical scales smaller than the depth of the main pycnocline and a life- time of one day (McWilliams and Molemaker, 2011; Lévy et al., 2012).

Despite a long history in studies of eddy activity, various aspects regarding processes and impacts of meso- and submesoscale eddies still remain unclear. One difficulty in the past and still today is the acquisition of a sufficient database to study these short-lived and small-scale phenomena, especially in the submesoscale regime. Satellite radar altimeter observes sea surface height (SSH); the difference between SSH and mean sea surface is known as sea surface height anomaly (SSHA), indicating the small displacement of sea surface elevation due to mesoscale eddies. For example, in anticyclonic eddies, the core is less dense (warm core) and has a high SSHA. On the contrary, in cyclonic eddies, the SSHA is lower and the density in the core is higher than in the surrounding area. Chlorophyll, suspended particulates, or other optically reflective materials in the water can reveal the motion in the visible channels (Robinson, 2010). In practice, an altimeter is commonly used for monitoring mesoscale eddies (Alpers et al., 2013), whereas submesoscale eddies are observed with infrared and optical sensors or SAR (Alpers et al., 2014). As the sea surface roughness could be modulated by eddies, SAR can also manifest signatures of eddies on an even smaller scale. Eddies can result in areas of surface convergence and divergence, which under moderate wind conditions appear

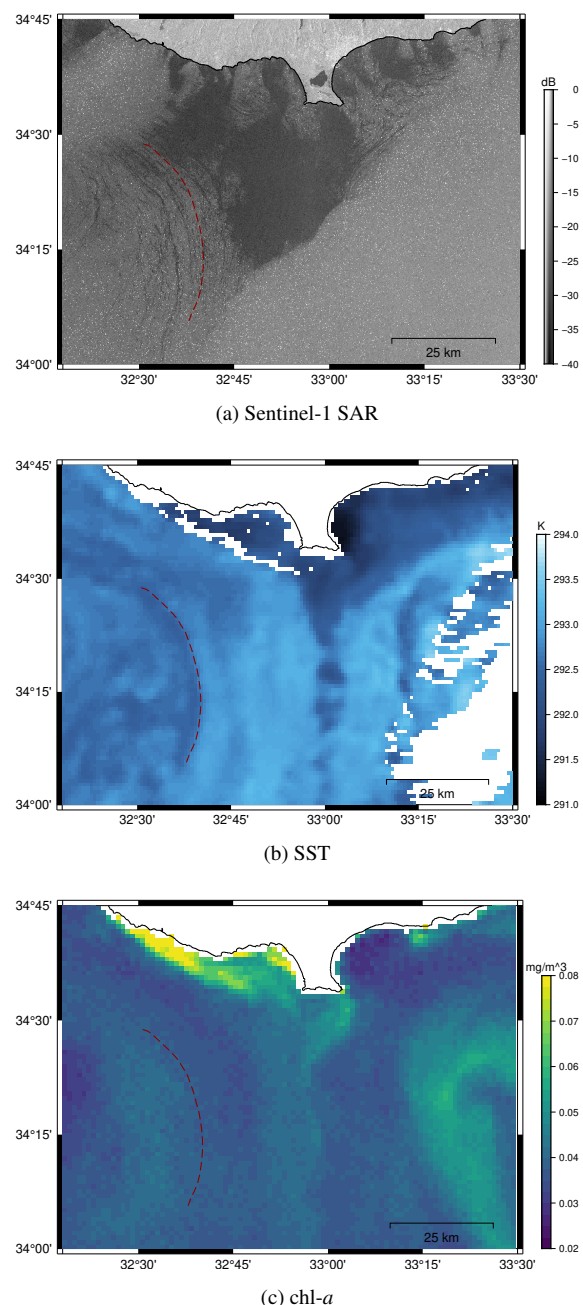

(a) Sentinel-1 SAR

(b) SST

(c) chl-*a*

**Figure 13.** An example of possible coastal upwelling in southern coast of Cyprus observed by (a) SAR at 03:51 on 12 May 2019 and the supporting materials, (b) SST and (c) chl-*a*, simulated at 00:00 on 12 May 2019 (CMEMS, a; CMEMS, b). The linear features to the left of the red dashed line in SAR indicates the possible surface films accumulations, which aligns well with the patterns of SST and chl-*a*. The figure contains modified Copernicus Sentinel data [2019]. The plotted coastline and borders were obtained from Wessel and Smith (1996).

as bright and dark lines in SAR images (Robinson, 2010). In addition, surface films tend to accumulate along the current shear in eddies. These films can dampen capillary waves and, in this way, enhance SAR signatures of the spiraling shear lines. They are sometimes regarded as "black" eddies in literature; on the other hand, "white" eddies refer to bright curved lines from eddies interacting with waves and current along the shear line at high wind speeds ranged between 7 and $12\,\mathrm{m\,s^{-1}}$ (Karimova and Gade, 2013).

Pegliasco et al. (2021) provided mesoscale eddy detection and tracking methods in the Mediterranean Sea based on products from radar altimeter data from 2000 to 2015 and gave an overview of eddy occurrence in the Mediterranean Sea; the Levantine basin is dominated by anticyclone eddies (warm core rings), which last longer than six months. In the southeastern Levantine basin, SAR observations indicated that the recurrent Cyprus and Shikmona eddies, located south of Cyprus at around $34°\,\mathrm{N}$ and west and east of $33.5°\,\mathrm{E}$, respectively, are dominant (Zodiatis et al., 2005; Gertman et al., 2007; Menna et al., 2012). However, the area is quite complex with cyclonic and anticyclonic eddies interacting with the alongshore cyclonic current and with each other (Gertman et al., 2007; Menna et al., 2012). Figure 14 illustrates such an example of mesoscale eddies forming spiral lines; the SAR scene was taken at 03:52 on 29 September 2022.

## 4.6 Biogenic Surface Film

There are two types of biogenic surface films that can reduce radar backscatter. Surface active organic molecules with hydrophobic and hydrophilic parts can form a molecular monolayer at the sea surface. The surface waves compress and dilate the molecular monolayer, leading to surface tension and surface potential gradients and thus generating the longitudinal Marangoni waves. The interaction of these waves and the transverse gravity capillary waves can result in Marangoni damping and reduce the radar backscattering (Hühnerfuss, 2006; Alpers et al., 2017). These surfactants usually originate from the wastewater or remnants of organisms in the water. Another type of biofilm consists of a much thicker layer with a high concentration of viscous floating materials such as phytoplankton, e.g., *Sargassum* and cyanobacteria (Qi et al., 2022). The accumulation of these floating materials decreases the surface tension of the water and dampens the gravity and capillary waves; this effect is similar to the effect of oil discussed in Subsect. 4.1.

Marine phytoplankton are ubiquitous in the sunlit layer of the oceans as they obtain energy through photosynthesis. Satellite-based studies have shown that the Mediterranean Sea generally has low chl-*a* concentration (a useful proxy for phytoplankton biomass), with concentrations often less than $0.2\,\mathrm{mg\,m^{-3}}$ with the exception of some blooming areas in the late winter and early spring (Siokou-Frangou et al., 2010). The overall chl-*a* concentration from satellite and in situ data shows a decline in the trends of west-to-east and north-to-south over the Mediterranean Sea (Siokou-Frangou et al., 2010). Even in blooming seasons, the chl-*a* concentration was rarely greater than $0.5\,\mathrm{mg\,m^{-3}}$ in the Eastern Mediterranean Sea (Siokou-Frangou et al., 2010), and similar concentrations have been reported from time-series in the Northwestern Mediterranean sea (von Jackowski et al., 2024).

Under a light wind condition, the convergent surface currents accumulate surface films (e.g. algae) along the current shear in fronts and eddies (Gade et al., 2013); these films can dampen the short gravity capillary waves and reduce SAR backscatter. On the other hand, the divergent surface currents make the films less concentrated and the dampening effect is less pro-

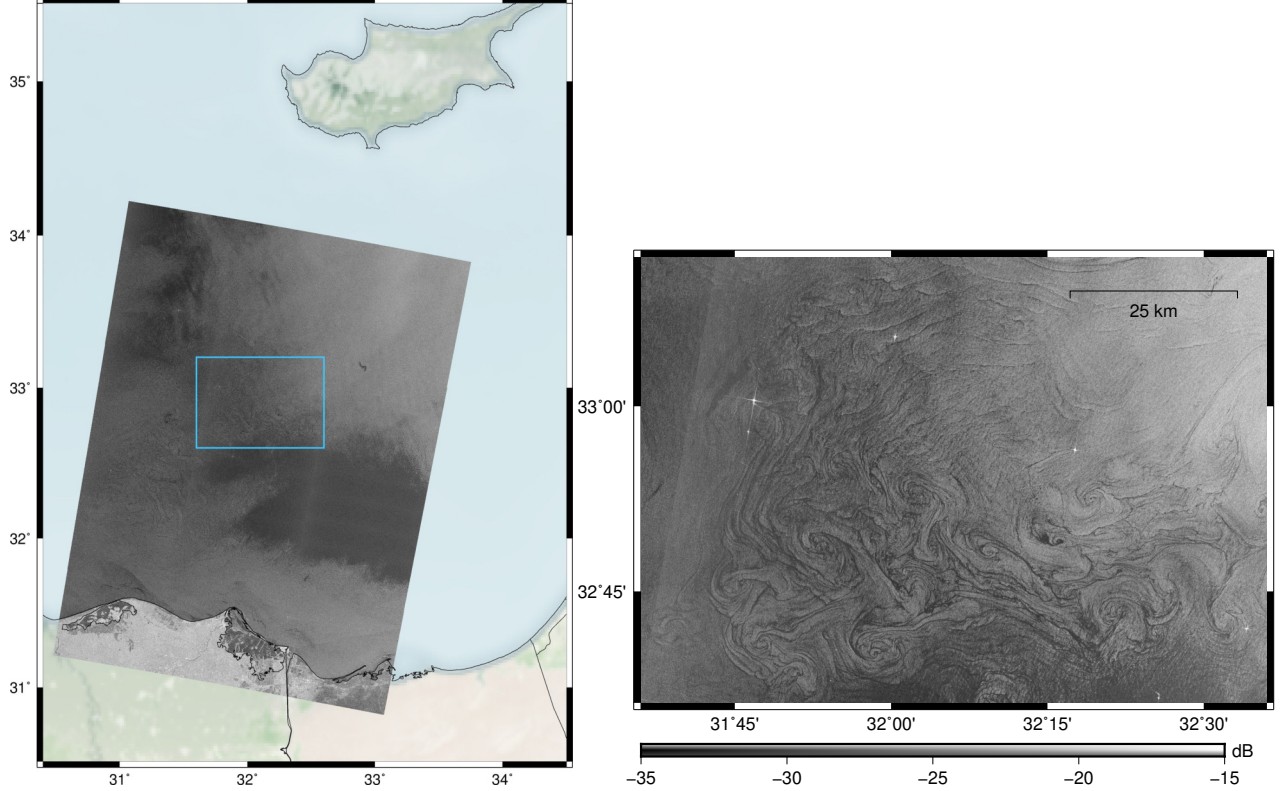

**Figure 14.** An example of mesoscale eddies observed by SAR at 03:52 on 29 September 2022. The right figure shows the zoomed-in area marked in blue in the left figure. The figure contains modified Copernicus Sentinel data [2022]. The base map of the left figure was obtained from Stevens (2020), and the coastline and borders were obtained from Wessel and Smith (1996).

nounced (Robinson, 2004). The chl-*a* concentration at the corresponding time and location of Figure 14 had a maximum of
$0.059 \, \mathrm{mg \, m^{-3}}$ and an average of $0.010 \, \mathrm{mg \, m^{-3}}$ (CMEMS, b), which suggests that the spiral patterns of eddies in Figure 14 were not due to accumulations of surface films. It is possible that the patterns were enhanced by a molecular monolayer, but this can not be confirmed without in situ water samples.

## 4.7   Rain cell

The rain-induced SAR signatures are contributed by a combination of surface scattering, volume scattering and attenuation
of radar pulse. Modulations of the sea surface roughness can come from several causes related to the rainfall. The impinging rain drops can either dampen or roughen the sea surface, leading to strong or weak backscattering. In addition, splash products from rain drops can cause scattering. Rain cells usually produce downward airflows (i.e., downdraft), which roughen the sea surface (Alpers et al., 2016). However, splash products from rain drops cause scattering and behind where the rain cell dropped on the sea surface in the direction of wind, the wave is damped (Atlas, 1994). Figure 15 shows such an example of

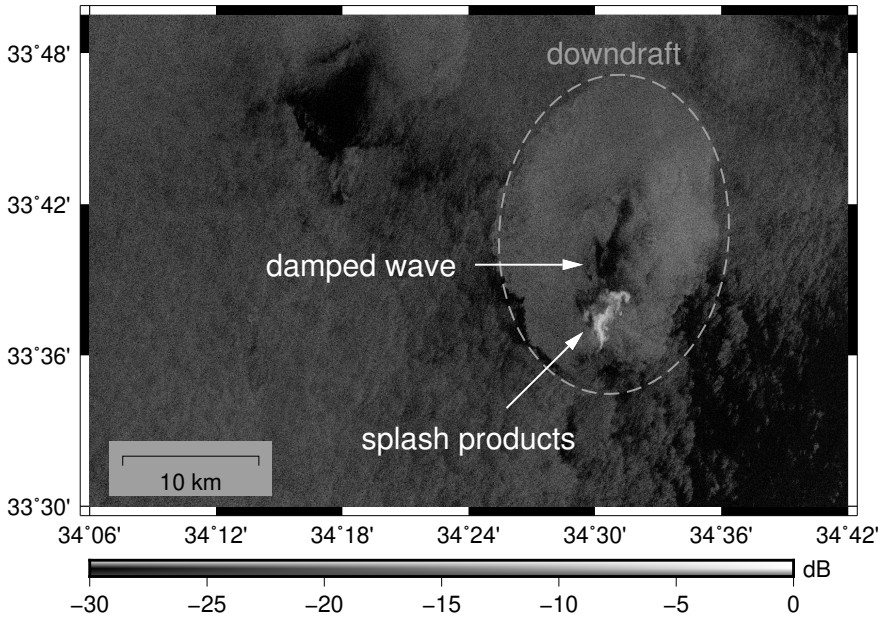

**Figure 15.** An example of a rain cell with downdraft patterns observed by SAR at 03:43 on 18 October 2022. The figure contains modified Copernicus Sentinel data [2022].

rain cell associated with downdraft, which roughened the sea surface and was shown as bright elliptical area. On top of that, hydrometeors also play important roles in the SAR signatures as they can cause volume scattering and attenuation of the radar pulse, which strengthen and weaken the radar backscattering, respectively. (Danklmayer et al., 2009; Alpers et al., 2016) At C-band, radar signatures for rain are complicated as the decrease or increase of the radar backscatter relative to the background is related to rain rate, wind speed, incidence angle, and time evolution of the rain event (Alpers et al., 2016). For low to moderate

high rain rate (smaller than $50$ mm hr$^{-1}$), the attenuation is negligible; for heavier rain, the attenuation can be greater than $1$ dB (Lin et al., 2001).

Figure 16 (a) shows a SAR scene covering the Israeli coast taken at 03:42 on 24 January 2018. The bright patches, located $5.5$ km to $15$ km away from the coast and quasi-parallel to it, indicate possible rain-induced signatures. To get an idea of the weather conditions in this region, daily and 10-minute rainfall measurements from the four coastal rain stations were obtained

from the Israel Meteorological Service (Ministry of Transport and Road Safety, Israel). The locations of these stations are plotted in Figure 16 (a), and their daily rainfall measurements on 24 January (one day in Israel time, UTC+2) are written in brackets. Though the automated 10-minute rainfall data tend to underestimate the rainfall in major events, they provide an overview of rainfall distribution for the day; Figure 16 (a) illustrates the 10-minute rainfall data from 00:00 to 07:00 on 24 January (UTC). The Hadera Port, Tel Aviv coast, and Ashdod Port stations all reported some rainfall between 02:30 and 05:30.

As the rain stations are not directly at the location of rain cells observed from SAR, rainfall at the coasts could be delayed or earlier than rainfall at the water.

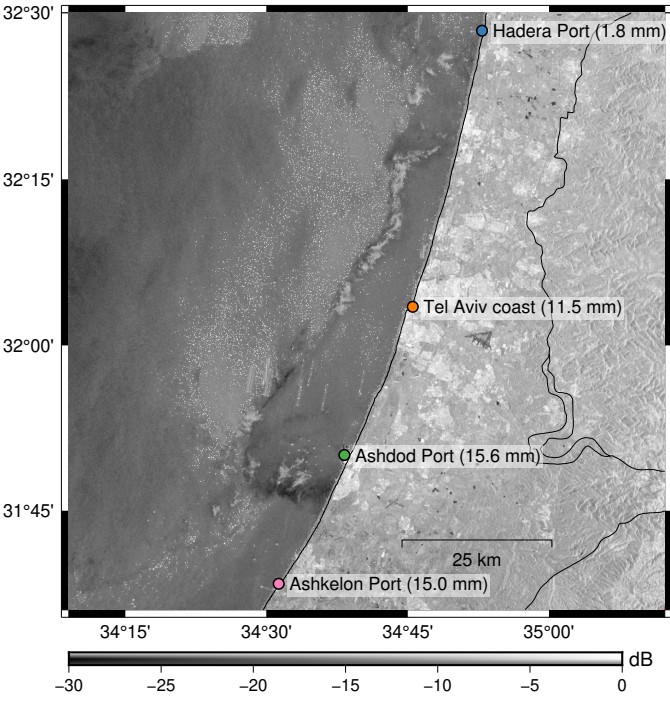

(a) Sentinel-1 SAR, location of rain stations, and the daily rainfall

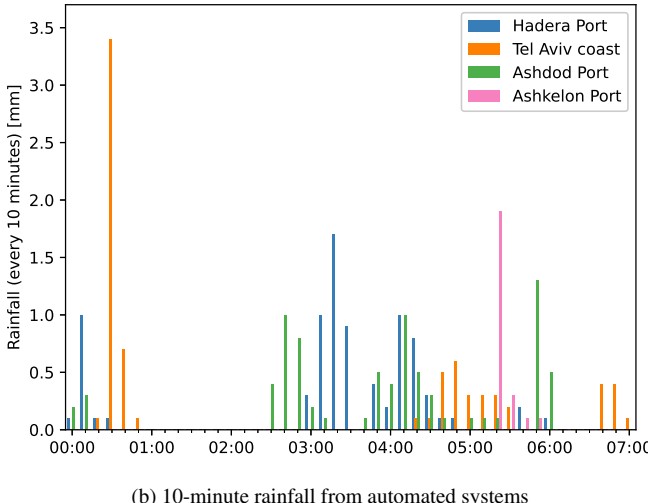

(b) 10-minute rainfall from automated systems

**Figure 16.** An example of rain cells observed by SAR at 03:42 on 24 January 2018, along with (a) daily and (b) 10-minute rainfall data obtained from the Israel Meteorological Service (Ministry of Transport and Road Safety, Israel). Figure (a) contains modified Copernicus Sentinel data [2018]. The coastline and borders were obtained from Wessel and Smith (1996).

## 4.8 Others

This section has explained oil slick and a variety of ocean and atmospheric phenomena as origins of SAR signatures. However, wakes and radio frequency interference (RFI), which are related to human activities, also cause remarkable SAR signatures. The following illustrates examples of those signatures, and related literature is provided to help users better understand them.

Moving vessels left tracks in the water as wakes; their structures can be categorized as surface waves (narrow-V wakes and Kelvin wake), turbulent wakes or vortices, and internal waves (Lyden et al., 1988). Although ships can be observed as bright patches in SAR image, ship wake patterns can provide further information such as size, direction, and speed (Rey et al., 1990); therefore, several previous studies focused on detection of ship wakes in SAR images (Lyden et al., 1988; Rey et al., 1990; Shemdin, 1990; Copeland et al., 1995; Graziano et al., 2017; Tings et al., 2023). Figure 17 shows a SAR scene taken at 03:52 on 18 October 2023 near the Port Said off the Egyptian coast. The bright pixels aligned in the south and north directions in the middle of the figure show vessels, along with dark linear features attached to them, which were likely due to wakes. Previous studies also pointed out that wakes from offshore wind turbines could form similar dark formations in SAR images (Christiansen and Hasager, 2005; Li and Lehner, 2013; Ahsbahs et al., 2020); however, they were not commonly seen in this study area.

As SAR is active radar, other radio services (such as communication systems, television networks, surveillance radars for air traffic control, military facilities, meteorological radars, and other spaceborne SAR sensors) with their transmitters on the same or adjacent frequency band to SAR could result in RFI, which usually appear as bright linear signatures in SAR (see Figure 18). Several previous studies discussed the influence of those RFI on different frequency bands and proposed removal methods (Miller et al., 1997; Rosen et al., 2008; Meyer et al., 2013; Natsuaki et al., 2017; Monti-Guarnieri et al., 2017; Franceschi et al., 2021). Thanks to researchers working on the detecting and identifying RFI in Sentinel-1 data, the operational RFI detection and mitigation was activated in the SAR processor on 23 March 2022 (Franceschi et al., 2022; Hajduch et al., 2022). In other words, since then, Sentinel-1 Level-1 data has been produced under consideration of RFI mitigation. However, readers should keep the effect of RFI in mind as they might still find some data with certain RFI not being mitigated; such examples are shown in Figure 19.

## 4.9 Example List

This subsection lists all Sentinel-1 SAR scenes used in the examples provided in this section. Some of the signatures covered a larger area than how they are shown in the paper; however, if readers wish to see the original SAR scenes, the information provided in Table 3 should be enough for obtaining them from the Copernicus Data Space Ecosystem. In addition, the corresponding supplementary materials used in the explanations are also listed; however, users should not limit themselves to the selection of supplementary data.

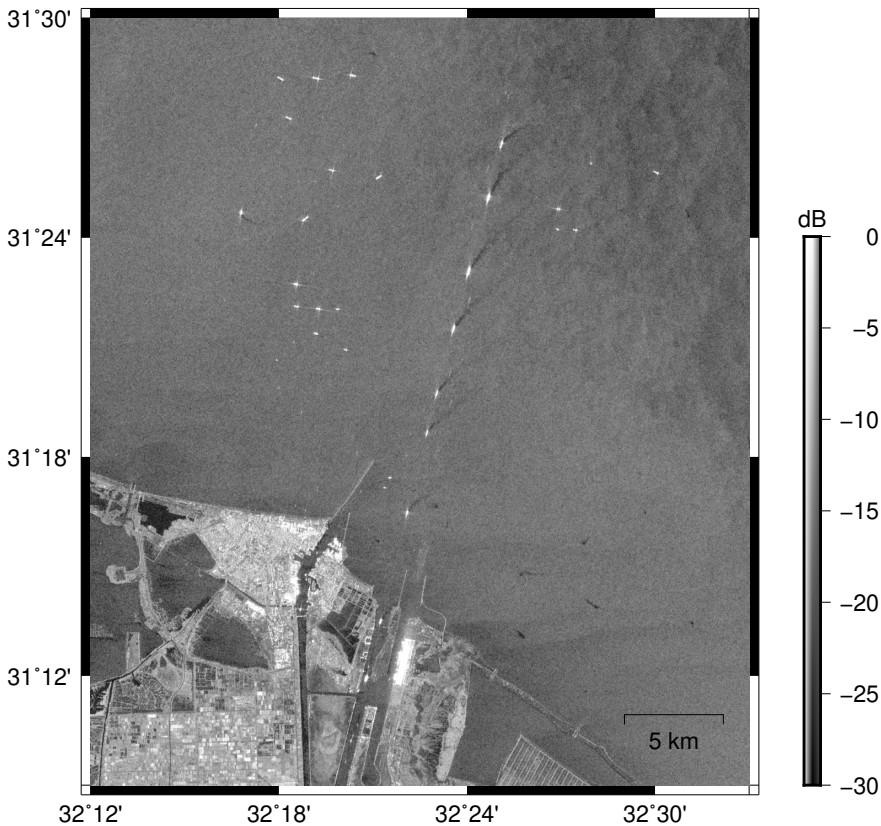

**Figure 17.** An example of ship wakes observed in a SAR scene taken at 03:52 on 18 October 2023 near the Port Said. The figure contains modified Copernicus Sentinel data [2023].

## 5 Usage Notes

### 5.1 Performance Evaluation

To make it possible for users who wish to compare their model performance with other studies, the performance of a custom-trained object detector from a previous study (Yang et al., 2024) was evaluated and is shown in this subsection. The performance on image patches with and without oil objects (i.e. *oil set* and *no-oil set*) are evaluated separately. In the following, the annotations from this published dataset are regarded as ground truth and abbreviated to *gt*. However, it shall be noted that even though the authors who prepared the dataset tried to avoid human errors, these annotations might still include false annotations. Note that this data descriptor is not intended to have a comprehensive discussion on the performance; therefore, this subsection only shows the measures, but readers can refer to Yang et al. (2024) for interpretation of the results.

For performance evaluation of object detection algorithms, intersection over union (IoU) is commonly used to indicate how accurate the detection (or known as prediction, which is a common term used in the object detection field) is compared to the

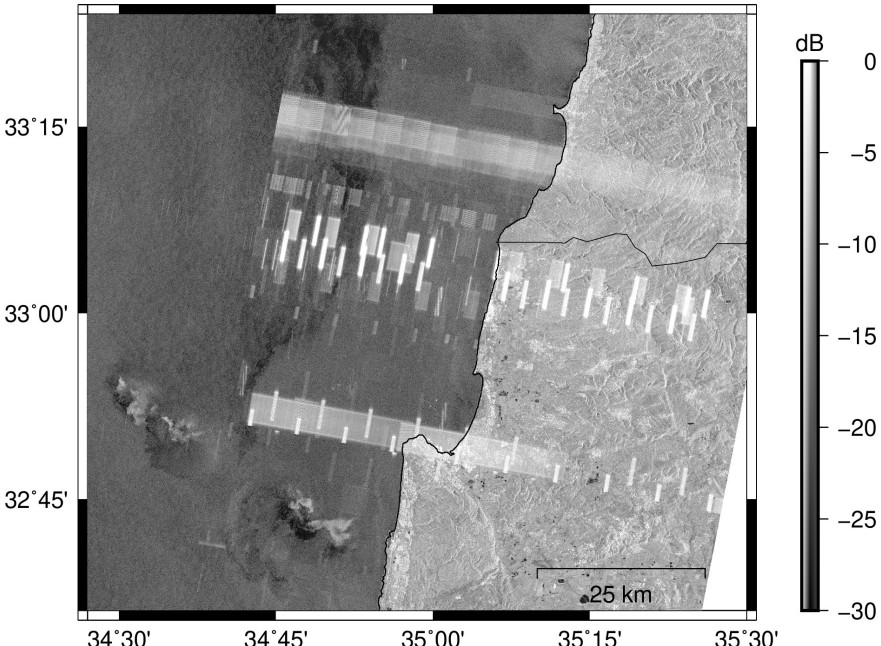

**Figure 18.** An example of RFI observed in a SAR scene taken at 03:43 on 21 October 2018, along with some rain cell signatures. The figure contains modified Copernicus Sentinel data [2018]. The plotted coastline and borders were obtained from Wessel and Smith (1996).

ground truth and is defined as (Everingham et al., 2010):

$$IoU = \frac{\text{area}(B_{detn} \cap B_{gt})}{\text{area}(B_{detn} \cup B_{gt})}, \tag{3}$$

where $B_{\text{detn}} \cap B_{\text{gt}}$ and $B_{\text{detn}} \cup B_{\text{gt}}$ refer to the intersection and union of the bounding boxes of the detection ($B_{\text{detn}}$) and the ground truth ($B_{\text{gt}}$), respectively.

Based on IoU, the detections can be categorized as true positives (TP) or false positives (FP). TP shows the detections intersecting with the ground truth and with their IoU greater than a given threshold. On the other hand, FP shows the detections with no intersection with the ground truth, or their IoU values are smaller than the threshold. If we change the perspective

and focus on ground truths, TP and false negatives (FN) are used, showing the ground truths with and without corresponding detections, respectively. The definition of one oil object might differ between ground truths and detections for some complicated cases, resulting in more than one detection referring to one ground truth. Therefore, the numbers of TP for detections and ground truths might differ; in the following, they are referred to as $\text{TP}_{\text{detn}}$ and $\text{TP}_{\text{gt}}$, respectively. To easily focus on the negative results of the models, the false discovery rate (FDR) and false negative rate (FNR) are provided, they are defined as follows:

$$FDR = \frac{FP}{TP_{detn} + FP},$$

$$FNR = \frac{FN}{TP_{gt} + FN}. \tag{4}$$

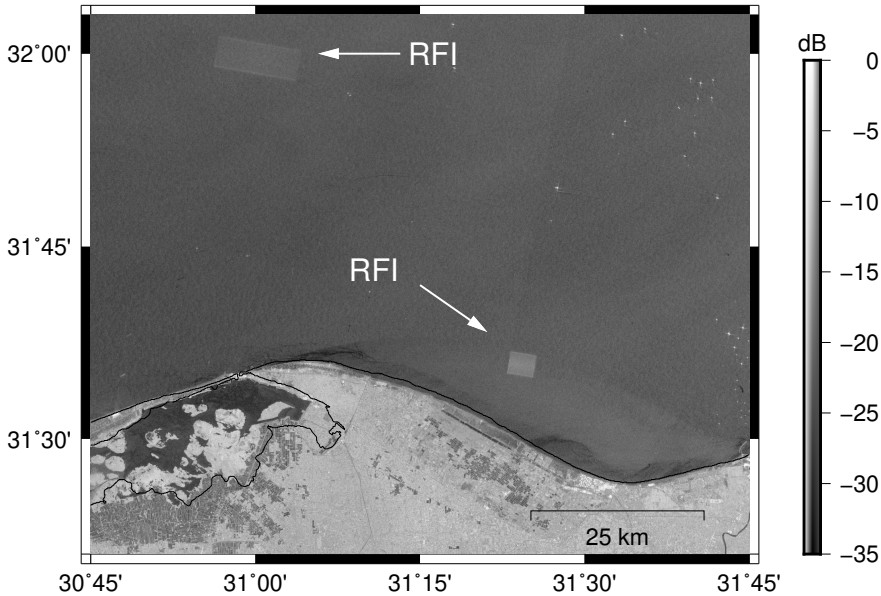

**Figure 19.** An example of RFI observed in a SAR scene taken at 03:52 on 24 September 2023; the RFI was not detected and mitigated by the SAR processor. The figure contains modified Copernicus Sentinel data [2023]. The plotted coastline were obtained from Wessel and Smith (1996).

Table 4 shows the performance of the two object detectors applied to a near real-time automated oil spill detection and early warning system as *YODA-enh* and *YODA-enh-aug1*, which were claimed to perform well in a previous study (Yang et al., 2024). Thresholds for filtering out the objects which have low confidence scores and IoU were applied. Note that due to different image patches used in the performance evaluation and a slightly different way of calculation, these numbers differ from those shown in the paper.

Regarding the performance on the *no-oil set*, as there are no objects inside, the measures for object detection algorithms are not appropriate. Here, image patches are simply marked as two categories, one with no detections inside and another with one or more detections inside. The former shows that the detector performs well and is not confused by the look-alikes. The latter indicates that the detector can be confused with specific signatures inside the image patches. Figure 20 shows the results of the two models on the *no-oil set* with different confidence score thresholds.

### 5.2 Technical Notes

There are some additional technical notes for the users:

– The annotation of the objects follows the Pascal VOC XML format; users who have their labels in different annotation format, should carefully convert the labels into the format they used.

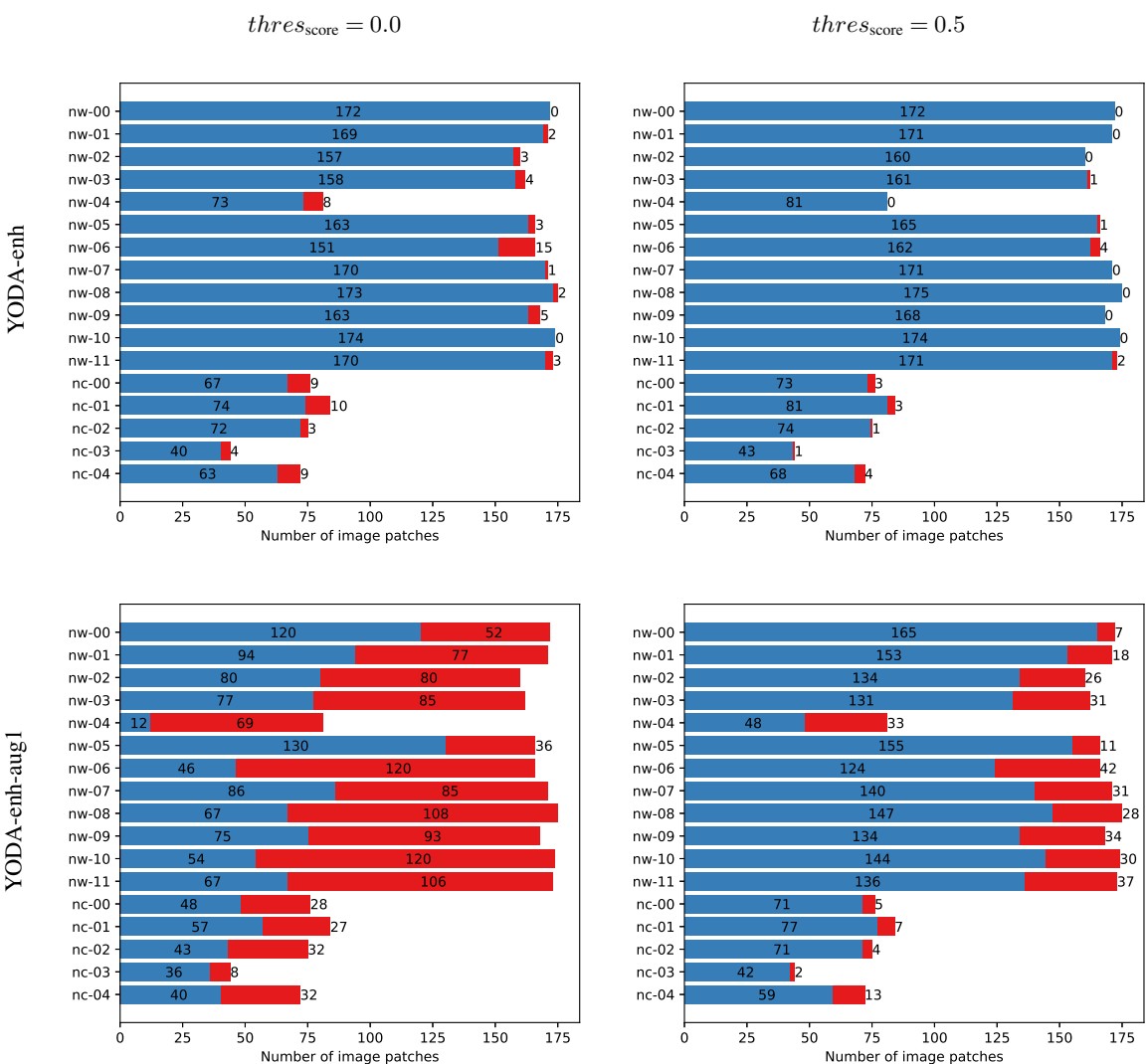

**Figure 20.** Numbers of image patches in the *no-oil set* that the models return detections (in red) and without any detection (in blue).

**Table 3.** List of Copernicus Sentinel-1 scenes used as examples provided in Sect. 4, along with their corresponding supplementary data.

| Fig. | Signature | Start Date & Time (YYYY-MM-DD hh:mm:ss) | | Abs. orbit | Mission ID | Product ID | Supplementary Data |
|---|---|---|---|---|---|---|---|
| 7 (a) | Oil slick | 2022-11-03 | 15:49:28 | 045732 | 057835 | D069 | |
| 7 (b) | Oil slick | 2022-11-04 | 03:52:42 | 045739 | 057870 | 6C96 | |
| 8 | Wind | 2023-08-02 | 03:44:34 | 049691 | 05F9AE | 7321 | • Sea surface wind velocity (CMEMS, c) |
| | | | 03:44:59 | | | D5EB | |
| 9 (a) | Wind | 2022-09-24 | 03:43:41 | 045141 | 056512 | 74D8 | • Sea surface wind velocity (CMEMS, c) |
| | | | 03:44:06 | | | F1AB | • Bathymetry and terrain chart (GEBCO, 2023) |
| 10 | Wind | 2023-01-27 | 03:52:39 | 046964 | 05A1FB | 3A5D | • Sea surface wind velocity (CMEMS, c) |
| 11 | Internal waves | 2024-03-28 | 15:41:26 | 053184 | 0671A3 | 05B8 | • Bathymetry and terrain chart (GEBCO, 2023) |
| | | | 15:41:51 | | | B6E5 | |
| 12 | AGWs | 2023-05-27 | 03:52:16 | 048714 | 05D8D7 | EF08 | • Sea surface wind velocity (CMEMS, c) |
| | | | 03:52:41 | | | 2FEB | |
| 13 | Upwelling | 2019-05-12 | 03:51:18 | 027189 | 0310AB | EEF7 | • SST (CMEMS, a) |
| | | | 03:51:43 | | | 00AD | • chl-*a* concentration (CMEMS, b) |
| | | | | | | | • Sea surface wind velocity (CMEMS, c) |
| 14 | Eddies | 2022-09-29 | 03:52:17 | 045214 | 056776 | 5889 | |
| | | | 03:52:42 | | | 04BC | |
| 15 | Rain cell | 2022-10-18 | 03:43:42 | 045491 | 057070 | 3FB9 | |
| | | | 03:44:07 | | | DC74 | |
| 16 | Rain cell | 2018-01-24 | 03:43:41 | 020291 | 022A5B | C066 | • Rainfall observation |
| | | | 03:44:06 | | | A779 | (Ministry of Transport and Road Safety, Israel) |
| 17 | Wakes | 2023-10-18 | 03:52:47 | 050814 | 061FD2 | 6D11 | |
| 18 | RFI | 2018-10-21 | 03:43:02 | 013245 | 0187C4 | ABBF | |
| 19 | RFI | 2023-09-24 | 03:52:47 | 050464 | 0613CC | CF0D | |

– The published dataset has only one class (i.e. oil) objects; if users have different definitions of classes in their work, they would have to make sure the performance evaluation is still valid.

   – For training an object detector, the authors would suggest using a dataset without masking out the land area as some coastal information might be helpful for the detectors to learn as background information. If users wish to apply land masks to the published dataset, they can generate the corresponding mask out scenes by loading the geoinformation

provided along with the dataset. A program for land masking is also provided in the GitHub repository associated with this paper.

**Table 4.** Performance evaluation of the models from a previous study (Yang et al., 2024) on the *oil set*.

| Model | | Subset | # img | # gt | # detn | # TP (gt) | # TP (detn) | # FP | # FN | FDR[%] | FNR[%] |
|---|---|---|---|---|---|---|---|---|---|---|---|
| YODA-enh | (a) | ow | 990 | 2284 | 1029 | 1154 | 1023 | 6 | 1249 | 0.58 | 51.98 |
| | | oc | 392 | 941 | 344 | 353 | 296 | 48 | 623 | 13.95 | 63.83 |
| | (b) | ow | 990 | 2284 | 468 | 515 | 468 | 0 | 1782 | 0.00 | 77.58 |
| | | oc | 392 | 941 | 137 | 148 | 130 | 7 | 798 | 5.11 | 84.36 |
| | (c) | ow | 990 | 2284 | 468 | 426 | 425 | 43 | 1858 | 9.19 | 81.35 |
| | | oc | 392 | 941 | 137 | 125 | 125 | 12 | 816 | 8.76 | 86.72 |
| YODA-enh-aug1 | (a) | ow | 990 | 2284 | 787 | 1487 | 787 | 0 | 877 | 0.00 | 37.10 |
| | | oc | 392 | 941 | 147 | 296 | 146 | 1 | 655 | 0.68 | 68.87 |
| | (b) | ow | 990 | 2284 | 318 | 520 | 318 | 0 | 1765 | 0.00 | 77.24 |
| | | oc | 392 | 941 | 55 | 101 | 55 | 0 | 842 | 0.00 | 89.29 |
| | (c) | ow | 990 | 2284 | 318 | 42 | 42 | 276 | 2242 | 86.79 | 98.16 |
| | | oc | 392 | 941 | 55 | 18 | 18 | 37 | 923 | 67.27 | 98.09 |

(a) $thres_{\mathrm{score}} = 0.0$, $thres_{\mathrm{IoU}} = 0.0$; (b) $thres_{\mathrm{score}} = 0.5$, $thres_{\mathrm{IoU}} = 0.0$; (c) $thres_{\mathrm{score}} = 0.5$, $thres_{\mathrm{IoU}} = 0.5$

– For an operational oil slick detection system, land masks should be considered as they can help improve the efficiency of the system and avoid false positives in the land areas. Global land masks can be obtained from sources such as Wessel and Smith (1996) and Karin (2020).

– For studies focusing on different study areas, the sources of oil slicks and geographic settings might be different from the dataset and, therefore, result in poor performance. However, it could help understand how one local model performs in a different area and lead to further discussion.

– Even though the authors aim to provide a published dataset as a test set for comparing models with different studies, users can also use the dataset to (further) train their algorithms.

– Users are encouraged to check the dataset themselves and adjust the labels when they do not fit the "style" of annotations from the users.

## 6   Summary

This data descriptor presents a dataset containing oil slicks, look-alikes, and other notable ocean phenomena, collected for the purpose of SAR image analysis. It provides explanations and examples of various ocean SAR signatures, supported by supplementary materials, to help users better understand the sources of these signatures.

The descriptor also includes a performance evaluation of a model from a previous study (Yang et al., 2024), which was tested on this dataset. This allows users who have trained their own oil spill detectors to compare their model performance with other studies. Additionally, the dataset is a valuable resource for newcomers to the oil slick detection community, providing a ready-made dataset for starting their work.

Recent research in SAR oil spill detection has increasingly focused on machine learning techniques. Given the diverse backgrounds of researchers in this field—ranging from remote sensing and machine learning to oceanography—this data descriptor aims to bridge the knowledge gap for those less familiar with oceanographic concepts. Readers are encouraged to consult specialized textbooks on SAR signal processing (Woodhouse, 2006), physical oceanography (Knauss and Garfield, 2016), and SAR oceanography (Robinson, 1983, 2004, 2010) for more comprehensive explanations.

## 7 Code and data availability

The dataset with oil slicks, look-alikes, and other remarkable phenomena covering the Eastern Mediterranean Sea in 2019 can be accessed through PANGAEA via https://doi.pangaea.de/10.1594/PANGAEA.980773 (Yang and Singha, 2025). The image patches are normalized to 0–255 and saved in an 8-bit JPG format. The oil annotations are in Pascal VOC XML format. A data table recording the Sentinel-1 ID of all the image patches inside the dataset is also provided; therefore, the original Sentinel-1 products can be downloaded via the Copernicus Data Space Ecosystem. The code related to working with the dataset, including the evaluation of performacne, is available on a public github repository under https://github.com/yi-jie-yang/dataset_DARTIS_2019 (last access: 21 November 2025).

## Appendix A: Data Table

Subsect. 2.2 provides explanations on how the dataset is organized and how their information is stored in the data table. To explain better and provide information about the image patches shown in this article, the data information of those image patches is extracted and displayed in the following tables; the original data table can be found in the published dataset. Tables A1 and A3 show the patch names, patch dimensions, and corner coordinates of the patches in the WGS84 geocentric coordinate system. Tables A2 and A4 are continuous from Tables A1 and A3, respectively, showing the product start and stop date time and the product ID. Table A5 contains the object information, including corner coordinates of the objects in the WGS84 geocentric coordinate system, object positions referring to the corresponding image patches along the range and azimuth direction, and the bounding box size in pixels. Tables A1 and A3 record the information of image patches from *no-oil set*, which are mentioned in Figures 2 and 6. On the other hand, Tables A3, A4, and A5 indicate the information of image patches from *oil set*, which are mentioned in Figure 5.

**Table A1.**

| tag | patch_name | patch_width | height | ul_lon | ul_lat | ur_lon | ur_lat | br_lon | br_lat | bl_lon | bl_lat |
|---|---|---|---|---|---|---|---|---|---|---|---|
| nw-0547-03-000044 | S1_20190319_035039_035218_VV_95 | 640 | 640 | 31.1062 | 33.3788 | 30.9706 | 33.3977 | 30.9475 | 33.2826 | 31.0829 | 33.2636 |
| nw-0553-03-000050 | S1_20190412_035040_035219_VV_12 | 640 | 640 | 32.3278 | 35.3925 | 32.1889 | 35.4121 | 32.1639 | 35.2971 | 32.3025 | 35.2774 |
| nw-0603-03-000100 | S1_20190803_035856_040035_VV_14 | 640 | 640 | 30.0576 | 35.9321 | 29.9176 | 35.9513 | 29.8933 | 35.8362 | 30.0331 | 35.8169 |
| nw-0609-03-000106 | S1_20190827_035858_040037_VV_21 | 640 | 640 | 29.9765 | 35.5871 | 29.8371 | 35.6063 | 29.8131 | 35.4911 | 29.9522 | 35.4719 |
| nw-0165-00-000165 | S1_20191220_035048_035227_VV_101 | 640 | 640 | 31.8779 | 34.4780 | 31.7405 | 34.4975 | 31.7165 | 34.3823 | 31.8537 | 34.3628 |
| nw-0307-01-000135 | S1_20191102_154017_154157_VV_101 | 640 | 640 | 33.9115 | 33.7039 | 34.0472 | 33.7251 | 34.0212 | 33.8404 | 33.8853 | 33.8192 |
| nw-0446-02-000103 | S1_20190711_035121_035303_VV_33 | 640 | 640 | 32.7978 | 35.3136 | 32.6592 | 35.3338 | 32.6336 | 35.2186 | 32.7720 | 35.1984 |
| nw-0651-03-000148 | S1_20191214_035125_035306_VV_294 | 640 | 640 | 31.8042 | 35.2799 | 31.6654 | 35.2990 | 31.6412 | 35.1839 | 31.7798 | 35.1648 |
| nw-0687-04-000022 | S1_20190406_035117_035258_VV_81 | 640 | 640 | 31.6319 | 34.9168 | 31.4937 | 34.9359 | 31.4704 | 34.8207 | 31.6084 | 34.8017 |
| nw-0763-05-000017 | S1_20190207_033454_033634_VV_22 | 640 | 640 | 35.6396 | 34.5452 | 35.5020 | 34.5642 | 35.4777 | 34.4492 | 35.6150 | 34.4302 |
| nw-0997-06-000085 | S1_20190522_155643_155737_VV_5 | 640 | 640 | 31.7808 | 32.6882 | 31.9154 | 32.7076 | 31.8919 | 32.8231 | 31.7572 | 32.8037 |
| nw-1203-07-000125 | S1_20190927_154017_154157_VV_86 | 640 | 640 | 34.0069 | 35.1136 | 34.1451 | 35.1343 | 34.1199 | 35.2495 | 33.9815 | 35.2289 |
| nw-1331-08-000082 | S1_20190531_034300_034440_VV_41 | 640 | 640 | 35.0884 | 35.5409 | 34.9495 | 35.5613 | 34.9242 | 35.4460 | 35.0630 | 35.4256 |
| nw-1523-09-000099 | S1_20190604_154838_155018_VV_33 | 640 | 640 | 32.7407 | 34.1620 | 32.8774 | 34.1821 | 32.8526 | 34.2972 | 32.7157 | 34.2772 |
| nw-1679-10-000087 | S1_20190821_035934_040114_VV_69 | 640 | 640 | 30.5790 | 31.6785 | 30.4463 | 31.6996 | 30.4210 | 31.5842 | 30.5535 | 31.5631 |
| nw-1883-11-000117 | S1_20190612_034300_034440_VV_90 | 640 | 640 | 35.2993 | 34.5986 | 35.1620 | 34.6195 | 35.1359 | 34.5043 | 35.2729 | 34.4834 |
| nc-0009-00-000009 | S1_20190216_035850_040029_VV_87 | 640 | 640 | 30.8020 | 31.5517 | 30.6695 | 31.5730 | 30.6439 | 31.4577 | 30.7763 | 31.4364 |
| nc-0140-01-000064 | S1_20191028_034227_034407_VV_119 | 640 | 640 | 35.5257 | 33.8044 | 35.3898 | 33.8258 | 35.3633 | 33.7105 | 35.4990 | 33.6891 |
| nc-0207-02-000047 | S1_20190927_035049_035228_VV_9 | 640 | 640 | 33.5185 | 34.7990 | 33.3810 | 34.8201 | 33.3556 | 34.7048 | 33.4929 | 34.6836 |
| nc-0253-03-000018 | S1_20190529_154748_154928_VV_50 | 640 | 640 | 32.2326 | 34.9748 | 32.3706 | 34.9952 | 32.3451 | 35.1106 | 32.2069 | 35.0903 |
| nc-0325-04-000046 | S1_20190909_035125_035306_VV_38 | 640 | 640 | 33.9199 | 34.9777 | 33.7821 | 34.9992 | 33.7550 | 34.8839 | 33.8925 | 34.8624 |

**Table A2.**

| tag | start_time | end_time | Sentinel_ID |
|---|---|---|---|
| nw-0547-03-000044 | 2019-03-19T03:50:39 | 2019-03-19T03:52:18 | S1B_IW_GRDH_1SDV_20190319T035104_20190319T035129_015418_01CDFB_86C8.SAFE |
| nw-0553-03-000050 | 2019-04-12T03:50:40 | 2019-04-12T03:52:19 | S1B_IW_GRDH_1SDV_20190412T035040_20190412T035105_015768_01D985_9F88.SAFE |
| nw-0603-03-000100 | 2019-08-03T03:58:56 | 2019-08-03T04:00:35 | S1B_IW_GRDH_1SDV_20190803T035856_20190803T035921_017416_020C16_EF88.SAFE |
| nw-0609-03-000106 | 2019-08-27T03:58:58 | 2019-08-27T04:00:37 | S1B_IW_GRDH_1SDV_20190827T035858_20190827T035923_017766_0216ED_C8C4.SAFE |
| nw-0165-00-000165 | 2019-12-20T03:50:48 | 2019-12-20T03:52:27 | S1B_IW_GRDH_1SDV_20191220T035048_20191220T035113_019443_024BA5_28B4.SAFE |
| nw-0307-01-000135 | 2019-11-02T15:40:17 | 2019-11-02T15:41:57 | S1A_IW_GRDH_1SDV_20191102T154107_20191102T154132_029734_036369_FE5F.SAFE |
| nw-0446-02-000103 | 2019-07-11T03:51:21 | 2019-07-11T03:53:03 | S1A_IW_GRDH_1SDV_20190711T035121_20190711T035146_028064_032B5E_8132.SAFE |
| nw-0651-03-000148 | 2019-12-14T03:51:25 | 2019-12-14T03:53:06 | S1A_IW_GRDH_1SDV_20191214T035125_20191214T035150_030339_037875_3C8E.SAFE |
| nw-0687-04-000022 | 2019-04-06T03:51:17 | 2019-04-06T03:52:58 | S1A_IW_GRDH_1SDV_20190406T035117_20190406T035142_026664_02FE02_C5A5.SAFE |
| nw-0763-05-000017 | 2019-02-07T03:34:54 | 2019-02-07T03:36:34 | S1A_IW_GRDH_1SDV_20190207T033454_20190207T033519_025818_02DF3B_8839.SAFE |
| nw-0997-06-000085 | 2019-05-22T15:56:43 | 2019-05-22T15:57:37 | S1B_IW_GRDH_1SDV_20190522T155643_20190522T155712_016359_01ECA7_4C33.SAFE |
| nw-1203-07-000125 | 2019-09-27T15:40:17 | 2019-09-27T15:41:57 | S1A_IW_GRDH_1SDV_20190927T154132_20190927T154157_029209_035141_93D0.SAFE |
| nw-1331-08-000082 | 2019-05-31T03:43:00 | 2019-05-31T03:44:40 | S1A_IW_GRDH_1SDV_20190531T034300_20190531T034325_027466_031954_5014.SAFE |
| nw-1523-09-000099 | 2019-06-04T15:48:38 | 2019-06-04T15:50:18 | S1A_IW_GRDH_1SDV_20190604T154928_20190604T154953_027532_031B5A_1FA3.SAFE |
| nw-1679-10-000087 | 2019-08-21T03:59:34 | 2019-08-21T04:01:14 | S1A_IW_GRDH_1SDV_20190821T040049_20190821T040114_028662_033E63_B88E.SAFE; S1A_IW_GRDH_1SDV_20190821T040024_20190821T040049_028662_033E63_3BDB.SAFE |
| nw-1883-11-000117 | 2019-06-12T03:43:00 | 2019-06-12T03:44:40 | S1A_IW_GRDH_1SDV_20190612T034325_20190612T034350_027641_031EA9_55D4.SAFE |
| nc-0009-00-000009 | 2019-02-16T03:58:50 | 2019-02-16T04:00:29 | S1B_IW_GRDH_1SDV_20190216T035940_20190216T040005_014966_01BF36_8C52.SAFE |
| nc-0140-01-000064 | 2019-10-28T03:42:27 | 2019-10-28T03:44:07 | S1B_IW_GRDH_1SDV_20191028T034252_20191028T034317_018670_0232F7_237B.SAFE |
| nc-0207-02-000047 | 2019-09-27T03:50:49 | 2019-09-27T03:52:28 | S1B_IW_GRDH_1SDV_20190927T035049_20190927T035114_018218_0224DC_62D9.SAFE |
| nc-0253-03-000018 | 2019-05-29T15:47:48 | 2019-05-29T15:49:28 | S1B_IW_GRDH_1SDV_20190529T154903_20190529T154928_016461_01EFC4_CD23.SAFE |
| nc-0325-04-000046 | 2019-09-09T03:51:25 | 2019-09-09T03:53:06 | S1A_IW_GRDH_1SDV_20190909T035125_20190909T035150_028939_034806_F874.SAFE |

**Table A3.**

| tag | patch_name | patch_width | height | ul_lon | ul_lat | ur_lon | ur_lat | br_lon | br_lat | bl_lon | bl_lat |
|---|---|---|---|---|---|---|---|---|---|---|---|
| ow-0267-01-000666 | S1_20190512_035118_035259_VV_1 | 640 | 640 | 31.4543 | 32.2251 | 31.3205 | 32.2448 | 31.2968 | 32.1295 | 31.4305 | 32.1099 |
| ow-0267-02-000667 | | | | | | | | | | | |
| ow-0267-03-000668 | | | | | | | | | | | |
| ow-0276-01-000680 | S1_20190515_160406_160546_VV_0 | 640 | 640 | 30.0586 | 31.4844 | 30.1915 | 31.5039 | 30.1681 | 31.6190 | 30.0351 | 31.5996 |
| ow-0276-02-000681 | | | | | | | | | | | |
| ow-0438-01-001124 | S1_20190622_035930_040110_VV_0 | 1294 | 1294 | 31.5372 | 33.7382 | 31.2640 | 33.7815 | 31.2107 | 33.5484 | 31.4840 | 33.5051 |
| ow-0438-02-001125 | | | | | | | | | | | |
| ow-0438-03-001126 | | | | | | | | | | | |
| ow-0438-04-001127 | | | | | | | | | | | |
| ow-0438-05-001128 | | | | | | | | | | | |
| ow-0447-01-001152 | S1_20190623_035044_035223_VV_0 | 977 | 977 | 32.8049 | 31.9612 | 32.6023 | 31.9935 | 32.5633 | 31.8176 | 32.7660 | 31.7852 |
| ow-0447-02-001153 | | | | | | | | | | | |
| ow-0447-03-001154 | | | | | | | | | | | |
| ow-0447-04-001155 | | | | | | | | | | | |
| ow-0447-05-001156 | | | | | | | | | | | |
| ow-0463-01-001189 | S1_20190624_034301_034441_VV_0 | 1748 | 1748 | 33.5068 | 32.5655 | 33.1398 | 32.6186 | 33.0742 | 32.3041 | 33.4412 | 32.2509 |
| ow-0463-02-001190 | | | | | | | | | | | |
| ow-0643-01-001540 | S1_20190821_154753_154933_VV_12 | 640 | 640 | 32.8779 | 31.8404 | 33.0111 | 31.8609 | 32.9864 | 31.9765 | 32.8532 | 31.9560 |
| ow-0643-02-001541 | | | | | | | | | | | |
| ow-0643-03-001542 | | | | | | | | | | | |
| ow-0643-04-001543 | | | | | | | | | | | |
| ow-0643-05-001544 | | | | | | | | | | | |
| ow-0643-06-001545 | | | | | | | | | | | |
| ow-0643-07-001546 | | | | | | | | | | | |
| ow-0643-08-001547 | | | | | | | | | | | |
| ow-0643-09-001548 | | | | | | | | | | | |
| ow-0643-10-001549 | | | | | | | | | | | |

**Table A4.**

| tag | start_time | end_time | Sentinel_ID |
|---|---|---|---|
| ow-0267-01-000666<br>ow-0267-02-000667<br>ow-0267-03-000668 | 2019-05-12T03:51:18 | 2019-05-12T03:52:59 | S1A_IW_GRDH_1SDV_20190512T035208_20190512T035233_027189_0310AB_E893.SAFE |
| ow-0276-01-000680<br>ow-0276-02-000681 | 2019-05-15T16:04:06 | 2019-05-15T16:05:46 | S1B_IW_GRDH_1SDV_20190515T160431_20190515T160456_016257_01E988_22CD.SAFE |
| ow-0438-01-001124<br>ow-0438-02-001125<br>ow-0438-03-001126<br>ow-0438-04-001127<br>ow-0438-05-001128 | 2019-06-22T03:59:30 | 2019-06-22T04:01:10 | S1A_IW_GRDH_1SDV_20190622T035955_20190622T040020_027787_0322FB_5391.SAFE |
| ow-0447-01-001152<br>ow-0447-02-001153<br>ow-0447-03-001154<br>ow-0447-04-001155<br>ow-0447-05-001156 | 2019-06-23T03:50:44 | 2019-06-23T03:52:23 | S1B_IW_GRDH_1SDV_20190623T035134_20190623T035159_016818_01FA62_D8E1.SAFE |
| ow-0463-01-001189<br>ow-0463-02-001190 | 2019-06-24T03:43:01 | 2019-06-24T03:44:41 | S1A_IW_GRDH_1SDV_20190624T034351_20190624T034416_027816_0323DB_253E.SAFE |
| ow-0643-01-001540<br>ow-0643-02-001541<br>ow-0643-03-001542<br>ow-0643-04-001543<br>ow-0643-05-001544<br>ow-0643-06-001545<br>ow-0643-07-001546<br>ow-0643-08-001547<br>ow-0643-09-001548<br>ow-0643-10-001549 | 2019-08-21T15:47:53 | 2019-08-21T15:49:33 | S1B_IW_GRDH_1SDV_20190821T154818_20190821T154843_017686_021461_4D57.SAFE |

**Table A5.**

| tag | obj_ ul_lon | ul_lat | ur_lon | ur_lat | br_lon | br_lat | bl_lon | bl_lat | obj_patchloc_ xmin | ymin | xmax | ymax | label_size |
|---|---|---|---|---|---|---|---|---|---|---|---|---|---|
| ow-0267-01-000666 | 31.3541 | 321779 | 31.3458 | 32.1791 | 31.3445 | 32.1728 | 31.3528 | 32.1716 | 420 | 334 | 460 | 369 | 1400 |
| ow-0267-02-000667 | 31.3592 | 322254 | 31.3475 | 32.2271 | 31.3427 | 32.2038 | 31.3544 | 32.2021 | 442 | 74 | 498 | 203 | 7224 |
| ow-0267-03-000668 | 31.4055 | 321367 | 31.3875 | 32.1394 | 31.3855 | 32.1297 | 31.4035 | 32.1270 | 142 | 515 | 228 | 569 | 4644 |
| ow-0276-01-000680 | 30.1084 | 315432 | 30.1230 | 31.5454 | 30.1197 | 31.5616 | 30.1051 | 31.5594 | 289 | 278 | 359 | 368 | 6300 |
| ow-0276-02-000681 | 30.1124 | 315529 | 30.1315 | 31.5557 | 30.1262 | 31.5820 | 30.1071 | 31.5792 | 317 | 327 | 409 | 473 | 13432 |
| ow-0438-01-001124 | 31.4387 | 336672 | 31.4193 | 33.6703 | 31.4186 | 33.6674 | 31.4381 | 33.6643 | 376 | 464 | 468 | 480 | 1472 |
| ow-0438-02-001125 | 31.4521 | 336606 | 31.4299 | 33.6641 | 31.4287 | 33.6589 | 31.4509 | 33.6554 | 308 | 488 | 413 | 517 | 3045 |
| ow-0438-03-001126 | 31.5102 | 336209 | 31.4874 | 33.6246 | 31.4856 | 33.6168 | 31.5085 | 33.6132 | 1 | 651 | 109 | 694 | 4644 |
| ow-0438-04-001127 | 31.4950 | 336443 | 31.4477 | 33.6518 | 31.4439 | 33.6354 | 31.4912 | 33.6279 | 095 | 539 | 319 | 630 | 20384 |
| ow-0438-05-001128 | 31.4419 | 337329 | 31.2973 | 33.7559 | 31.2836 | 33.6959 | 31.4282 | 33.6730 | 430 | 109 | 1115 | 442 | 228105 |
| ow-0447-01-001152 | 32.7687 | 317987 | 32.7480 | 31.8020 | 32.7450 | 31.7885 | 32.7657 | 31.7852 | 1 | 902 | 101 | 977 | 7500 |
| ow-0447-02-001153 | 32.7912 | 319593 | 32.7507 | 31.9657 | 32.7317 | 31.8796 | 32.7721 | 31.8732 | 062 | 22 | 257 | 500 | 93210 |
| ow-0447-03-001154 | 32.7728 | 318173 | 32.7670 | 31.8182 | 32.7633 | 31.8015 | 32.7691 | 31.8005 | 1 | 799 | 29 | 892 | 2604 |
| ow-0447-04-001155 | 32.7840 | 318726 | 32.7440 | 31.8790 | 32.7319 | 31.8244 | 32.7719 | 31.8180 | 6 | 493 | 199 | 796 | 58479 |
| ow-0447-05-001156 | 32.7246 | 319242 | 32.7111 | 31.9264 | 32.7062 | 31.9042 | 32.7197 | 31.9020 | 336 | 267 | 401 | 390 | 7995 |
| ow-0463-01-001189 | 33.4992 | 325301 | 33.2166 | 32.5710 | 33.1687 | 32.3412 | 33.4513 | 32.3003 | 1 | 197 | 1347 | 1474 | 1718842 |
| ow-0463-02-001190 | 33.5066 | 325654 | 33.2281 | 32.6057 | 33.1786 | 32.3678 | 33.4570 | 32.3275 | 1 | 1 | 1327 | 1323 | 1752972 |
| ow-0643-01-001540 | 32.9035 | 318446 | 32.9484 | 31.8515 | 32.9432 | 31.8758 | 32.8983 | 31.8689 | 123 | 1 | 339 | 136 | 29160 |
| ow-0643-02-001541 | 32.9057 | 318665 | 32.9335 | 31.8708 | 32.9133 | 31.9652 | 32.8854 | 31.9610 | 155 | 117 | 289 | 640 | 70082 |
| ow-0643-03-001542 | 32.9400 | 318755 | 32.9473 | 31.8767 | 32.9427 | 31.8985 | 32.9354 | 31.8974 | 324 | 137 | 359 | 258 | 4235 |
| ow-0643-04-001543 | 32.9266 | 319080 | 32.9312 | 31.9087 | 32.9265 | 31.9309 | 32.9219 | 31.9302 | 294 | 322 | 316 | 445 | 2706 |
| ow-0643-05-001544 | 32.9370 | 319609 | 32.9605 | 31.9645 | 32.9590 | 31.9717 | 32.9355 | 31.9681 | 395 | 597 | 508 | 637 | 4520 |
| ow-0643-06-001545 | 32.9340 | 318996 | 32.9374 | 31.9001 | 32.9357 | 31.9079 | 32.9324 | 31.9074 | 320 | 271 | 336 | 314 | 688 |
| ow-0643-07-001546 | 32.9215 | 319322 | 32.9231 | 31.9324 | 32.9218 | 31.9386 | 32.9202 | 31.9383 | 294 | 456 | 302 | 490 | 272 |
| ow-0643-08-001547 | 32.9191 | 319372 | 32.9220 | 31.9377 | 32.9211 | 31.9420 | 32.9182 | 31.9416 | 288 | 485 | 302 | 509 | 336 |
| ow-0643-09-001548 | 32.9164 | 319508 | 32.9247 | 31.9521 | 32.9232 | 31.9591 | 32.9149 | 31.9578 | 289 | 560 | 329 | 599 | 1560 |
| ow-0643-10-001549 | 32.9188 | 319424 | 32.9213 | 31.9428 | 32.9196 | 31.9507 | 32.9171 | 31.9504 | 292 | 513 | 304 | 557 | 528 |

*Author contributions.* YJY and SS jointly inspected the oil slicks in SAR scenes. YJY prepared the dataset and wrote the manuscripts. SS
supported with his knowledge of SAR techniques. RG and FS contributed with valuable feedback on the explanations of oceanographic and
atmospheric phenomena. All authors reviewed the manuscript.

*Competing interests.* No potential conflict of interest was reported by the author(s).

*Acknowledgements.* The dataset generation was partly funded by the DARTIS project, supported by the German Federal Ministry of Education and Research under grant number 03F0823B. The authors wish to thank Copernicus Programme for providing Sentinel-1 data,
Copernicus Marine Service, and the other Earth observation data. The authors would also like to thank C. Pegel, C. Schnupfhagn, and D.
Günzel from the German Aerospace Center and P. Brandt from GEOMAR for their valuable feedback on the manuscripts. Additionally, the
authors would like to thank the reviewers for their helpful comments.

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
