# Peer review of "Dataset of Oil Slicks, Look-Alikes and Remarkable SAR Signatures Obtained from Sentinel-1 Data in the Eastern Mediterranean Sea"

_Earth System Science Data, 2025_

## Author Response (AR1)

**Comments from Referee 2**

The detection of oil slicks in the real ocean is crucial for effective natural disaster management using satellite technology, such as Synthetic Aperture Radar (SAR). However, the complexity of oceanic phenomena makes real-world oil slick identification challenging. This difficulty is partly due to the lack of a robust benchmark dataset suitable for training effective detection models. The manuscript by Yang et al. attempts to address this critical need by providing a categorized dataset. I recommend the publication of this manuscript following the incorporation of the suggestions below.

1. Data Hosting and Model Interoperability: Is it possible to establish a website to host not only the data but also the model codes, similar to initiatives like the PIV challenge (see https://www.pivchallenge.org/)? This would facilitate fair comparison between different models and ensure ease of use for the wider research community.

   Thanks for the advice. The codes have now been released on GitHub: https://github.com/yi-jie-yang/dataset_DARTIS_2019. The GitHub link has also been now included in the section "Code and data availability". Since the code has been provided, the functions listed in Appendix A1 have been removed.

2. Terminology (Line 47): Please replace "vertical advection in the ocean" with the more precise oceanographic term: "upwelling in the ocean."

   Revised.

3. Equation (1): Please provide a brief justification for choosing the parameter threshold as three times the standard deviation.

   There is no specific function that should be applied for image normalization. The parameters used to generate the dataset were determined by the authors through trial and error, as they provide good contrast between oil slicks and their surroundings according to human eyes. The object detectors used in Subsect. 5.1 were trained with image patches to which a different image normalization method was applied. Therefore, different image normalization methods may not be the key factor in the poor performance of the object detection algorithms. Explanations regarding this point are now included in the revised manuscripts after Equation (1).

4. Annotation Criteria (Lines 111-112): Regarding the data labeled "jointly by two human interpreters," please comment on the specific criteria used for achieving consensus between the interpreters.

   The oil slicks were labeled by two human interpreters, only the dark formations agreed upon by both interpreters as oil slicks were given labels (included in the revised manuscripts).

5. Figure 3 Caption: The caption should be more informative. For example, the meaning of the vertical line must be explicitly mentioned, even though it is referenced in the main text.

   The revised caption is as follows: Numbers of image patches in different clusters by K-Mean algorithm. The vertical red lines indicate the number of image patches randomly selected from the subsets. These image patches were then manually inspected, and the number of image patches in the final published dataset from each subgroup is presented in Table 1.

6. Clustering Justification (Lines 189-190): Regarding the "K-means clustering methods with 12 and 5 classes for nw and nc subsets," please provide further detail or comment on the rationale for selecting 12 and 5 classes, respectively.

The purpose of using the K-Means clustering method is to provide a "no-oil set" in which the signatures of different phenomena appear more balanced. In other words, the clustering method categorized the dataset, and then the same number of image patches from each class were randomly selected. However, for better understanding the performance of the object detector on different kinds of SAR signatures, it would be ideal if all image patches within each class have similar sources of phenomena that could be explained by humans. Therefore, human interpretation was considered when selecting the number of classes, which was adjusted by reviewing image patches in different classes. In addition, the following were considered when deciding the different numbers of classes for "water" (12) and "coast" (5) subsets: 1. The total number of image patches in "nw" subset is much more than the one in "nc" subset. 2. Sea states in coastal areas are often complicated by factors such as bathymetry and interaction with land. However, open water typically experiences larger-scale dynamics that can manifest a variety of different SAR signatures. Explanations are now included in the Subsect. 3.2.

7. Data Pre-processing (Line 325): When geolocation information is available, the removal of land and islands could be easily and efficiently handled using the global-land-mask Python package (https://github.com/toddkarin/global-land-mask). This should be considered for data cleaning.

For training an object detector, the authors would suggest using a dataset without masking out the land area as some coastal information might be helpful for the detectors to learn as background information. However, for an operational oil slick detection system, land masks should be considered as they can help improve the efficiency of the system and avoid false positives in the land areas. If users wish to apply land masks to the published dataset, they can generate the corresponding mask out scenes by loading the geoinformation provided along with the dataset. A program for land masking is also provided in the GitHub repository associated with this paper. The above explanation along with some suggested sources of land masks have been included in the Subsect. 5.2 as an additional point for technical notes.

8. Figure 13: It would significantly improve clarity to visually indicate the upwelling area in all sub-figures.

The example used in Figure 13 has been changed. The caption for this example should provide a sufficient description of the upwelling area.

9. Typographical Error (Line 33): Please remove the redundant "in" in the sentence.

The redundant "in" in Line 433 has been removed.

10. Notation Consistency (Equation (3)): Ensure the notation used for the IoU (Intersection over Union) in Equation (3) is consistent with its use elsewhere in the main text.

For better consistency with Equation (4), the notations used for IoU are revised to use "detn" for detections and "gt" for ground truths.

11. Model Identification (Line 559): Please explicitly provide the names of the "two models" being referenced.

The names are now included: Table 4 shows the performance of the two object detectors were applied to a near real-time automated oil spill detection and early warning system as YODA-enh and YODA-enh-aug1, which were claimed to perform well in a previous study (Yang et al., 2024).

**Additional revisions**

In addition to the revisions related to the referees' comments, several other revisions were made:

1. The authors identified some errors relating to the plotting of wind speed and direction on the maps. These errors have now been corrected in the revised manuscripts.

2. The explanations of streak patterns in the last paragraph of Section 2 could lead to a misunderstanding that Langmuir circulation is only related to wind. To avoid confusion, the paragraph has been revised.

3. The example provided in Figure 13 was not ideal as the wind speeds in this area were quite low (0–3 m/s). The signatures were more likely to come from low wind speed than upwelling. Therefore, a new example is provided in the revised manuscripts.

4. Figure 11 was intended to illustrate oceanic internal waves (OIWs); however, the authors also received comments from colleagues suggesting that the dark and bright strips could be caused by atmospheric gravity waves (AGWs). Therefore, the explanations relating to the figure were revised.

---

## Author Response (AR2)

Minor revision on wind speed mentioned in Subsect. 4.1.